# Measuring the Interpretability of Unsupervised Representations via Quantized Reverse Probing

**Iro Laina**
University of Oxford
iro.laina@eng.ox.ac.uk

**Yuki M. Asano**
University of Amsterdam
y.m.asano@uva.nl

**Andrea Vedaldi**
University of Oxford
vedaldi@robots.ox.ac.uk

## Abstract

Self-supervised visual representation learning has recently attracted significant research interest. While a common way to evaluate self-supervised representations is through transfer to various downstream tasks, we instead investigate the problem of measuring their interpretability, *i.e.* understanding the semantics encoded in raw representations. We formulate the latter as estimating the mutual information between the representation and a space of manually labelled concepts. To quantify this we introduce a decoding bottleneck: information must be captured by simple predictors, mapping concepts to clusters in representation space. This approach, which we call *reverse linear probing*, provides a single number sensitive to the semanticity of the representation. This measure is also able to detect when the representation contains combinations of concepts (*e.g.*, "red apple") instead of just individual attributes ("red" and "apple" independently). Finally, we propose to use supervised classifiers to automatically label large datasets in order to enrich the space of concepts used for probing. We use our method to evaluate a large number of self-supervised representations, ranking them by interpretability, highlight the differences that emerge compared to the standard evaluation with linear probes and discuss several qualitative insights. Code at: https://github.com/iro-cp/ssl-qrp.

## 1 Introduction

Relying on black-box models such as deep networks comes sometimes with significant methodological and ethical challenges. This is particularly true for *unsupervised* and *self-supervised* models which are learned without human supervision. While these models perform increasingly well in downstream applications, often outperforming supervised counterparts, there is very little understanding of what they learn, making their real-life deployment risky.

In this paper, we thus consider the problem of characterizing the *meaning* of data representations, with a particular focus on unsupervised and self-supervised representations of images. Given a representation $f$ mapping images $x$ to representation vectors $f(x) \in \mathbb{R}^D$, our goal is to find whether these vectors contain human-interpretable information. This is usually done by finding a relationship or correspondence between $f(x)$ and human-provided descriptions $y(x)$ of the images, essentially translating the information from the representation space to a concept space. Consider for example the popular linear probing method (Alain and Bengio, 2017). Given a large dataset $\hat{\mathcal{X}}$ of images with corresponding manual annotations $y(x)$, one learns linear classifiers (*probes*) to map the feature vectors $f(x)$ to labels $y(x)$, measuring the resulting classification accuracy. If the predictor is accurate, then one can argue that the representation captures the corresponding concept $y$.

One possible issue with this type of approaches is that one can only discover in the representation concepts that are represented in the available data annotations. In order to maximize the semantic coverage of the analysis, it is thus customary to combine annotations for several types of attributes, such as object classes, textures, colour, etc. (Bau et al., 2017). In this case, $y$ is a vector of image attributes, and one can compute several linear probes to predict each individual attribute $y_i$. By doing so, however, attributes are treated independently, which may be unsatisfactory. In order to obtain a single number assessing the overall semanticity of a representation, one must combine

the prediction accuracies of several independent classifiers, and there is no natural way of doing so (*e.g.*, a simple average would not account for the different complexity of the different prediction tasks). Furthermore, the representation might be predictive of *combinations* of attributes, *i.e.* it might understand the concept of "red apple" without necessarily understanding the individual concepts, "red" and "apple". While it is in principle possible to test any combination of attributes via linear probing, most attribute combinations are too rare to generate significant statistics for this analysis.

We propose a complementary assessment strategy to address these shortcomings. In contrast to linear probes, we start by considering the reverse prediction problem, mapping label vectors $y(x)$ to representation vectors $f(x)$. A key advantage is that the entire attribute vector is used for this mapping, which, as we show later, accounts for attribute combinations more effectively.

Next, we consider the challenge of deriving a quantity that allows to compare representations based on the performance of these reverse predictors. Obviously a simple metric such as the average $\mathcal{L}_2$ prediction error is meaningless as its magnitude would be affected by irrelevant factors such as the scale of the representation vectors. To solve this problem in a principled manner, we consider instead the *mutual information* between the concepts and quantized representation vectors. This approach, which we justify from the viewpoint of information theory, essentially measures whether the representation groups the data in a human-interpretable manner.

We use our approach to evaluate a large number of self-supervised and supervised image representations. Importantly, we show that **(a)** all methods capture interpretable concepts that extend well beyond the underlying label distribution (*e.g.*, ImageNet labels, which are not used for training), **(b)** while some clusters that form in the representation space are purely semantic (object-centric), others carry information about scenery, material, textures, or *combined* concepts, and **(c)** context matters. We also show that more performant methods recover more of the original label distribution, *i.e.* learn better "ImageNet concepts", and rely less on lower-level concepts. Quantitatively, we observe that our interpretability measure results in a similar but not identical ranking for state-of-the-art methods with clustering-based approaches generally producing more interpretable representations.

## 2 RELATED WORK

**Self-supervised representation learning (SSL)** In this paper, we focus on self-supervised methods that learn representations from image data. Early self-supervised learning approaches devise numerous pretext tasks — colorization, predicting image rotations or solving image puzzles — to learn useful representations from unlabeled images (Gidaris et al., 2018; Pathak et al., 2016; Noroozi and Favaro, 2016; Doersch et al., 2015; Larsson et al., 2016; Zhang et al., 2017; 2016). More recent approaches follow the contrastive learning paradigm (Chen et al., 2020a; Frankle et al., 2020; He et al., 2020; Chen et al., 2020b; Hénaff et al., 2019; Oord et al., 2018; Misra and Maaten, 2020; Tian et al., 2019; 2020; Wu et al., 2018), where the goal is to discriminate instances belonging to the same image from negative samples. Later methods take a closer look at the effect of negatives samples (Mitrovic et al., 2020; Robinson et al., 2020; Chuang et al., 2020; Kalantidis et al., 2020) or eliminate the need for negatives altogether (Zbontar et al., 2021; Chen and He, 2020; Grill et al., 2020; Caron et al., 2021), while others consider nearest neighbors (Dwibedi et al., 2021; Assran et al., 2021). Although several studies aim to explain why constrastive learning works (Arora et al., 2019; Wang and Isola, 2020; Tschannen et al., 2020; Tsai et al., 2020; Purushwalkam and Gupta, 2020), in most cases, the focus lies on measuring empirical improvements in downstream tasks.

**Clustering** A related part of literature deals with unsupervised image clustering. Early approaches include using autoencoders (Hinton and Salakhutdinov, 2006), agglomerative clustering (Bautista et al., 2016) and partially ordered sets of hand-crafted features (Bautista et al., 2017). More recent methods combine representation learning with clustering, using mutual information (Ji et al., 2019; Hu et al., 2017), $K$-means (Caron et al., 2018; Zhan et al., 2020), prototypes (Li et al., 2021) or optimal transport (Asano et al., 2020; Caron et al., 2020). Other methods build on strong feature representations and, at a second stage, cluster and further refine the network (Yan et al., 2020; Van Gansbeke et al., 2020; Lee et al., 2019). These methods produce pseudo-labels which can be used to evaluate representations by measuring the correlation to the ground-truth labels.

**Evaluation and analysis of SSL** There are two main directions in evaluating self-supervised learning methods. The first one uses pre-trained representations as an initialization for subsequent

supervised tasks, *e.g.*, for object detection. The second approach is to train a linear classifier on the representation space which is kept frozen after pre-training. Some concerns have been voiced regarding the generality of ImageNet linear classification accuracy as the main metric to evaluate representations (Kotar et al., 2021) and, as a result, several benchmarking suites with various datasets and tasks have been proposed (Zhai et al., 2019; Goyal et al., 2019; Van Horn et al., 2021; Kotar et al., 2021). Complementary to this, clustering in the frozen feature space has also been proposed as an evaluation metric (Zheltonozhskii et al., 2020; Sariyildiz et al., 2020). In addition, downstream dataset (Ericsson et al., 2021) and pretraining dataset (Zhao et al., 2021; Cole et al., 2021) dependencies have been evaluated. Finally, recent investigations of self-supervised feature spaces aim to understand separability (Sehwag et al., 2020), concept generalization (Sariyildiz et al., 2020) and the effect of balanced vs. long-tailed training data (Kang et al., 2020).

**Interpretability**  Although a large amount of work has studied feature representations in Convolutional Neural Networks (CNNs), it is heavily focused on models trained with full supervision. Zeiler and Fergus (2014) and Zhou et al. (2014) analyze networks by visualizing the most activating patches, while activation maximization methods (Mahendran and Vedaldi, 2016a;b; Nguyen et al., 2017; 2016; Olah et al., 2017; Simonyan et al., 2014) generate inputs to activate specific neurons.

Another line of work focuses on understanding what information is present in intermediate representations of CNNs, usually mapping activations to high-level concepts. Alain and Bengio (2017) introduce linear probes as a means to understanding the dynamics of intermediate layers, by predicting the target labels from these layers. Escorcia et al. (2015) also use a linear formulation to study the relationship of mid-level features and visual attributes, while Oramas et al. (2019) adopt this to predict task-specific classes from the features (similar to Alain and Bengio (2017)), select relevant features and generate a visual explanation. Similarly, Zhou et al. (2018) decompose feature vectors into a set of elementary and interpretable components and show decomposed Grad-CAM heat maps (Selvaraju et al., 2017) for concepts that contribute to the prediction. Furthermore, Kim et al. (2018) relate feature vectors and human-interpretable concepts using a set of user-provided examples for each concept and Bau et al. (2017) propose to quantify the interpretability of individual units by measuring the overlap between each unit and a set of densely annotated concepts, Finally, Ghorbani et al. (2019) propose a method to automatically assign concepts to image segments and Yeh et al. (2020) analyze the completeness of these concepts for explaining the CNN.

With the exception of (Laina et al., 2020; Bau et al., 2017; Fong and Vedaldi, 2018), relatively little work has been done to understand the emergence of interpretable visual concepts in self-supervised representations specifically. In particular, Laina et al. (2020) quantify the learnability and describability of unsupervised image groupings, by measuring how successful humans are at understanding the learned concept from a set of provided examples. Instead, our approach does not depend on human input and is thus easily scalable to a wide range of methods and number of classes.

## 3  METHOD

We are interested in measuring the semanticity of data representations. A representation $f$ is a map $f : \mathcal{X} \to \mathbb{R}^D$ that encodes data samples $x \in \mathcal{X}$ as vectors $f(x) \in \mathbb{R}^D$. In this paper, we assume that the data are images $\mathcal{X} = \mathbb{R}^{3 \times H \times W}$ and $f$ is a deep neural network, but other choices are possible.

The semantic content of an image $x$ can be summarized by obtaining a label or description $y(x) \in \mathcal{Y}$ of the image from a human annotator. If the representation $f(x)$ captures the meaning of the image well, then it should be predictive of the description $y(x)$. The mutual information $I(f(x), y(x))$ provides a natural measure of predictivity, and it is thus tempting to use this quantity as a measure of the overall semanticity of the representation. Note that, due to the data processing inequality $(I(f(x), y(x)) \leq I(x, y(x)))$, the information is maximized by observing the raw image, *i.e.* by the identity representation $f(x) = x$. Since data processing cannot increase information content, a useful representation must preserve information while *also* making it easier to decode and act on it.

There are many possible definitions of what constitutes "easy decoding". Common in literature is the transfer through a simple predictor (*e.g.*, linear) to new tasks. Here we propose a definition that stems from the interpretation of differential entropy as the limit of discrete entropy for quantized versions of the corresponding variables (Cover and Thomas, 2006). In our case, the description $y(x)$ is already discrete, but the representation vectors are continuous. We thus propose to discretize (vector-

Figure 1: Overview of our approach. (1) we evaluate pre-trained SSL models on an image collection, extracting and quantizing feature vectors to obtain clusters ( ▪▪▪▪·▪ ), (2) we label the image data with a diverse set of concepts ( ▪▪▪▪▪ ···▪▪ ) from expert models trained with supervision on external data sources, and (3) we train a linear model $h_\theta$ to map concepts to clusters, measuring the mutual information between the representation and human-interpretable concepts.

quantize) the representation and to use its information as a measure of semanticity. Intuitively, in the limit of infinitely-fine quantization, this quantity reduces (up to a normalizing term) to the mutual information above. For a finite number of clusters, however, a large value of information means that the representation groups images in a way that makes sense to a human observer. In our case, the quantization amounts to grouping images based on the $L^2$-distance between their representation vectors, *i.e.* clustering.

Formally, we use a vector quantization algorithm such as $K$-means (Lloyd, 1982) to learn a quantizer function $\mu_K : \mathbb{R}^D \to \{1, \dots, K\}$ for the representation vectors, and estimate the mutual information $I(\mu_K(f(x)), y(x))$ by rewriting it as:

$$I(f_K(x), y(x)) = H(f_K(x)) - H(f_K(x) \mid y(x)), \quad f_K(x) = \mu_K(f(x)). \tag{1}$$

In the above equation, the first term denotes the entropy of the cluster assignments. In practice, given a sample dataset $\mathcal{X}$ of images, we first run $K$-means to compute the quantizer $\mu_K$, and then compute the frequency of cluster assignments $f_K(x)$, $x \in \mathcal{X}$ to calculate the entropy. The second term in Eq. (1) is the conditional entropy:

$$H(f_K(x) \mid y(x)) = \mathbb{E}_p[-\ln p(f_K(x) \mid y(x))] \leq \mathbb{E}_p[-\ln q(f_K(x) \mid y(x))],$$

where $p$ is the true joint distribution between variables $f_K(x)$ and $y(x)$. While this is difficult to compute, we can upper-bound it by considering an auxiliary posterior distribution $q$ interpreting the conditional entropy as the cross-entropy loss of a *predictor* $h_\theta : \mathcal{Y} \to K$, parametrized by $\theta$, that maps labels $y(x)$ to clusters $f_K(x)$. The gap can be minimized by learning the parameters $\theta$.

In practice, learning the predictor may result in overfitting, which would lead to an over-optimistic estimate of the mutual information. We address this issue by learning the predictor $h_\theta$ on a subset $\hat{\mathcal{X}} \subset \mathcal{X}$ of the data and evaluating its cross-entropy loss on the remaining subset $\mathcal{X} - \hat{\mathcal{X}}$. Importantly, we consider a linear predictor (*probe*) for $h_\theta$, to further reduce the risk of overfitting. We refer to this as *quantized reverse probing*, as it maps semantic concepts to quantized data representations.

**Number of clusters** The number of clusters $K$ controls the size of the bottleneck. Mutual information increases monotonically as $K$ increases, as we show in the Appendix (Fig. 6); when comparing representations, it is thus important to fix a value of $K$ and use it consistently. For added convenience, in practice we report a normalised value of information (NMI).

**Obtaining labelled data via automatic experts** Similar to prior work on interpretation, our method requires a dataset $x \in \mathcal{X}$ of images equipped with manually-provided labels $y(x) \in \mathcal{Y}$. Because we do not know *a-priori* which concepts may be captured by a given representation $f(x)$ under analysis, these labels should provide a good coverage for a number of different concepts (*e.g.*, textures, object types, parts, etc.). A good example of such a dataset is Broden (Bau et al., 2017).

Because it would be cumbersome, and maybe even infeasible, to annotate any target dataset with *all* present visual concepts, a cost-effective way to increase the coverage of the label space is to *predict* the labels $y(x)$ automatically via a battery of expert classifiers learned using full supervision. The noise of these automated predictions is small compared to the noise in the correlation between the concepts and the unsupervised/self-supervised visual representations under study.

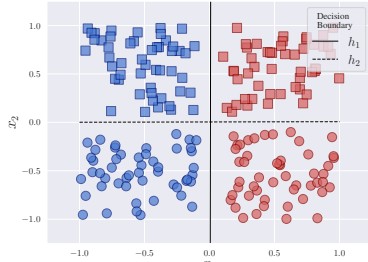 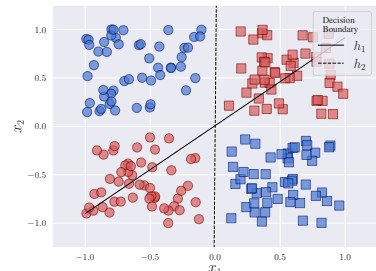

(a) All attributes are linearly separable (Acc[$h_1$] = Acc[$h_2$] = 100%).

(b) Color attribute not linearly separable (Acc[$h_1$] = 50%, Acc[$h_2$] = 100%).

Figure 2: **Forward vs. reverse probing.** A dataset $\mathcal{X}$ embedded in $\mathbb{R}^2$ by two different representations, with meaningful clusters in representation space. Each data point has two binary attributes, **color** $y_1(x)$: red or blue and **shape** $y_2(x)$: □ or ○. Both representations separate these attributes; however, color is not linearly separable in (b), so forward linear probing cannot recover this relationship. Decision boundaries are shown for the forward linear probes. On the contrary, our reverse probing easily discovers that all *combinations* of shape and color map to well separated clusters.

**Relation to linear probing**   Linear probing is a common approach for assessing representations. A linear probe is a simple predictor $h_i \circ f(x) \approx y_i(x)$ that maps the representation $f(x)$ to a specific concept $y_i$ (*e.g.*, a binary attribute or a class label). The idea is that the representation captures concept $y_i$ if the probe performs well. As a result, linear probing has become a standard evaluation protocol in self-supervised representation to measure the (linear) classification accuracy of the learned representations against a labelled set, commonly ImageNet (Russakovsky et al., 2015). In interpretability studies, different formulations of linear probing (Alain and Bengio, 2017) are used to understand intermediate layers of neural networks. Similar to our approach, the simplicity of the probes (imposed via linearity and sometimes sparsity) is a necessary bottleneck for interpretability: without such a constraint, the best possible probe would take the raw image as input.

Our method is complementary to linear probes, with some advantages. To discuss this point further, we show an example in Fig. 2, where data points with color and shape attributes are encoded by two different representations, such that they form well-defined *nameable* clusters, *i.e.* blue square, red square, blue circle, and red circle. Forward linear probes ($\mathcal{X} \rightarrow \mathcal{Y}$) can be trained as binary classifiers $h_1$, $h_2$ for each attribute, mapping a feature vector (in this case, coordinates) to the corresponding attribute value for each data point. If attributes are linearly separable (Fig. 2a), then forward probes do well at separating the representation space. If an attribute is not linearly separable (color in Fig. 2b), the predictive accuracy of the corresponding probe $h_1$ reduces to chance, and consequently reduces the average score. As a result, linear probes cannot always assess the meaningfulness of clusters or, in other words, whether clusters in representation space respond to well-defined concepts. On the contrary, if all data points in a cluster have consistent attributes, the reverse probe ($\mathcal{Y} \rightarrow \mathcal{X}$) achieves a high score for, *e.g.*, "red square" without requiring the concept "red" to be *also* encoded in the features. In this case, we can say that the model has discovered the concept of a "red square" without identifying, in isolation, the concept "red". Therefore, reverse probing handles combinations of attributes naturally and allows to better assess the semanticity of clusters.

## 4   EXPERIMENTS

We use our method to evaluate a wide range of recent self-supervised representation learning and clustering techniques. In the following, we compare these techniques according to the proposed criterion, which results in a different ranking for state-of-the-art approaches. Further, we show qualitatively how our method discovers elementary concepts encoded in the raw representations.

### 4.1   IMPLEMENTATION DETAILS

**Attributes and expert models**   As a first step in our approach, we collect a set of attributes which we expect to find encoded in the self-supervised representations; these include semantic object cat-

Table 1: Different types of concepts used in our evaluation and corresponding datasets.

| Categories | Tasks/Datasets |
|---|---|
| OBJECTS | IN-1k (Russakovsky et al., 2015); Open Images (Kuznetsova et al., 2020); MSCOCO [things] (Lin et al., 2014) |
| SCENE | Places-365 (Zhou et al., 2016); Scene attributes (Patterson and Hays, 2012) |
| MATERIAL | MINC (Bell et al., 2015); MSCOCO [stuff] (Caesar et al., 2018) |
| TEXTURE | DTD (Cimpoi et al., 2014); Color (Bau et al., 2017) |
| OTHER | Text detection; Sentiment; Photographic style |

egories, scene types and attributes, materials and textures and possibly other information about the photographic style or overall sentiment of an image (Table 1). We thus look for relevant human-annotated datasets and expert models trained on these datasets, as a proxy to human knowledge. This allows us to extract information related to such attributes on a different dataset or images in the wild; in our experiments we focus on ImageNet (IN-1k) (Deng et al., 2009) as the target data due to the large availability of pre-trained self-supervised models on this dataset. We concatenate all available attributes and form $M$-dimensional binary vectors denoting the presence or absence of each attribute in an image. We provide full details about the expert models in Appendix A. Since we primarily focus on ImageNet, it is natural to also use the existing 1000 human-annotated labels. As IN-1k already includes several fine-grained categories, we have not included further experts from common fine-grained recognition datasets, such as iNaturalist (Van Horn et al., 2018).

**Linear model**  For each method, we train a linear model mapping a set of input attributes to clusters, which are obtained by $K$-means clustering on top of fixed representations and evaluate on a held-out set. Further training details are provided in Appendix C.

## 4.2 COMPARISON OF SELF-SUPERVISED REPRESENTATIONS

We use reverse probing to evaluate recent self-supervised methods and rank them based on their semanticity (Table 2). Following convention, we categorize these models as contrastive (⊕⊖), positive-only, (⊕), clustering-based (⊞), and handcrafted pretexts (⚒) — details are provided in Appendix B. For each method, we freeze the pre-trained model and extract feature vectors on the IN-1k train data. We run $5\times K = 1000$-means to obtain different clusterings, each used to train a linear model using *all* categories from Table 1. In addition to a normalized measure of mutual information (NMI), we also report the adjusted mutual information (AMI), classification accuracy (top-1) and mean average precision (mAP). We also note the linear classification accuracy, as reported by the respective source for each method. For fairness of comparisons, we group methods based on the number of training epochs. For reference, we also evaluate the supervised counterparts in the same way as the self-supervised methods. Overall, we find a relatively strong correlation between our approach and the common linear evaluation protocol (Fig. 3). We also make the following observations:

**Ranking**  Despite the strong correlation to linear classification accuracy, we obtain a different ranking from the viewpoint of representational interpretability, using our approach. In particular, state-of-the-art methods DINO, SwAV and BYOL fall slightly behind in our evaluation. The most recent version of MoCo (v3) performs the best across three metrics, with DeepCluster-v2 and OBoW ranking in the second place (across ResNet-50-based models).

**Number of pre-training epochs**  For some methods, learned weights are available at various epochs. In terms of interpretability, we observe no benefit in longer pre-training in the case of DeepCluster-v2 and SwAV, despite the solid performance gains on standard benchmarks. However, there is a noticeable gain for MoCo-v2, MoCHi and SimCLR between 200 and 800 epochs. Also notable is that one of the best performing methods, OBoW is only trained for 200 epochs.

**Clustering vs contrastive approaches**  Methods such as SeLa, OBoW, PCL and DeepCluster-v2, which use a clustering mechanism during training, have generally more interpretable representations, *i.e.* they rank higher than methods with similar linear classification accuracy.

**Architecture**  Interpretability increases with self-supervised methods that use a ViT (Touvron et al., 2021; Dosovitskiy et al., 2021) backbone, showing a significant boost in all metrics, even more so than what linear classification accuracy suggests. However, a fair comparison between MoCo-v3 and DINO is not possible due to pre-training for a different number of epochs (300 vs. 800).

Table 2: Evaluation of self-supervised learning methods on ImageNet-1k (Russakovsky et al., 2015). Lin. Acc. refers to the linear classification accuracy reported by the respective source for each model. The remaining metrics are computed for our approach and reported as mean ($\pm\sigma$) over 5 clusterings.

| | Model | | Lin. Acc. | NMI | AMI | Top-1 | mAP |
|---|---|---|---|---|---|---|---|
| | **ResNet-50** ($1\times$) | | | | | | |
| Epoch 200 | Jigsaw | (Goyal et al., 2019) | 46.58 | 37.36 ± 0.04 | 22.17 ± 0.04 | 9.80 ± 0.04 | 5.95 ± 0.04 |
| | ClusterFit | (Yan et al., 2020) | 53.63 | 51.10 ± 0.07 | 38.78 ± 0.10 | 30.06 ± 0.09 | 25.41 ± 0.11 |
| | CMC | (Tian et al., 2019) | 58.60 | 51.96 ± 0.06 | 39.33 ± 0.07 | 29.01 ± 0.15 | 25.77 ± 0.23 |
| | MoCo-v1 | (He et al., 2020) | 60.60 | 55.94 ± 0.05 | 44.39 ± 0.07 | 34.36 ± 0.13 | 33.36 ± 0.17 |
| | SeLa-v1 | (Asano et al., 2020) | 61.50 | 59.91 ± 0.04 | 49.02 ± 0.08 | 41.22 ± 0.13 | 39.94 ± 0.30 |
| | SimCLR | (Chen et al., 2020a) | 66.61 | 63.73 ± 0.07 | 54.87 ± 0.09 | 47.93 ± 0.09 | 53.95 ± 0.08 |
| | MoCo-v2 | (Chen et al., 2020b) | 67.70 | 64.76 ± 0.05 | 56.05 ± 0.05 | 47.53 ± 0.10 | 51.97 ± 0.19 |
| | InfoMin | (Tian et al., 2020) | 70.10 | 65.16 ± 0.07 | 57.27 ± 0.05 | 48.11 ± 0.11 | 54.79 ± 0.13 |
| | MoCHi | (Kalantidis et al., 2020) | 67.60 | 65.27 ± 0.09 | 56.94 ± 0.13 | 49.26 ± 0.22 | 56.15 ± 0.40 |
| | PCL-v2 | (Li et al., 2021) | 67.60 | 68.89 ± 0.06 | 62.13 ± 0.09 | 53.72 ± 0.08 | 61.26 ± 0.16 |
| | SwAV | (Caron et al., 2020) | 73.90 | 69.18 ± 0.05 | 61.68 ± 0.06 | 55.85 ± 0.09 | 62.77 ± 0.27 |
| | OBoW | (Gidaris et al., 2021) | 73.80 | **71.50** ± **0.07** | 64.09 ± 0.09 | 57.25 ± 0.13 | 61.67 ± 0.22 |
| Epoch 400 | SimCLR | (Chen et al., 2020a) | 67.71 | 64.88 ± 0.09 | 56.45 ± 0.12 | 50.03 ± 0.17 | 56.92 ± 0.21 |
| | SwAV | (Caron et al., 2020) | 74.60 | 69.09 ± 0.12 | 61.66 ± 0.18 | 56.06 ± 0.22 | 62.78 ± 0.25 |
| | SeLa-v2 | (Asano et al., 2020) | 71.80 | 70.04 ± 0.06 | 63.43 ± 0.07 | 58.47 ± 0.15 | **68.33** ± **0.34** |
| | DeepCluster-v2 | (Caron et al., 2020) | 74.32 | 70.85 ± 0.07 | 64.01 ± 0.12 | 58.91 ± 0.15 | 67.60 ± 0.27 |
| Epoch 800 | SimCLR | (Chen et al., 2020a) | 69.68 | 65.63 ± 0.14 | 57.48 ± 0.15 | 51.36 ± 0.17 | 59.11 ± 0.31 |
| | PIRL | (Misra and Maaten, 2020) | 69.90 | 65.92 ± 0.05 | 58.79 ± 0.09 | 51.95 ± 0.07 | 60.75 ± 0.11 |
| | DINO | (Caron et al., 2021) | 75.30 | 68.73 ± 0.08 | 61.18 ± 0.11 | 55.33 ± 0.21 | 59.87 ± 0.21 |
| | SwAV | (Caron et al., 2020) | 75.30 | 68.79 ± 0.05 | 61.19 ± 0.08 | 55.73 ± 0.06 | 62.17 ± 0.28 |
| | MoCHi | (Kalantidis et al., 2020) | 69.20 | 69.00 ± 0.07 | 61.77 ± 0.06 | 55.23 ± 0.07 | 63.81 ± 0.09 |
| | MoCo-v2 | (Chen et al., 2020b) | 71.10 | 69.02 ± 0.07 | 61.55 ± 0.11 | 54.17 ± 0.17 | 60.44 ± 0.10 |
| | InfoMin | (Tian et al., 2020) | 73.00 | 69.20 ± 0.02 | 62.54 ± 0.04 | 55.15 ± 0.10 | 63.84 ± 0.10 |
| | DeepCluster-v2 | (Caron et al., 2020) | 75.18 | 69.44 ± 0.04 | 62.12 ± 0.05 | 57.01 ± 0.08 | 63.97 ± 0.26 |
| Epoch 1k | Barlow Twins | (Zbontar et al., 2021) | 73.50 | 69.37 ± 0.08 | 61.69 ± 0.13 | 56.84 ± 0.18 | 61.32 ± 0.23 |
| | MoCHi | (Kalantidis et al., 2020) | 70.60 | 70.16 ± 0.06 | 63.37 ± 0.09 | 57.08 ± 0.16 | 66.18 ± 0.11 |
| | BYOL | (Grill et al., 2020) | 74.40 | 70.48 ± 0.07 | 63.12 ± 0.10 | 58.36 ± 0.09 | 63.26 ± 0.25 |
| | MoCo-v3 | (Chen et al., 2021) | 74.60 | **71.45** ± **0.06** | **64.49** ± **0.09** | **59.58** ± **0.11** | 64.95 ± 0.43 |
| | Supervised | (He et al., 2016) | − | 83.20 ± 0.06 | 78.82 ± 0.07 | 76.16 ± 0.14 | 78.53 ± 0.15 |
| | **ViT-Base/16** | | | | | | |
| | MoCo-v3 | (Chen et al., 2021) | 76.70 | 79.06 ± 0.04 | 73.67 ± 0.05 | 70.51 ± 0.09 | 74.39 ± 0.28 |
| | DINO | (Caron et al., 2021) | 78.20 | 81.46 ± 0.08 | 76.70 ± 0.11 | 72.95 ± 0.14 | 76.44 ± 0.18 |
| | Supervised | (Touvron et al., 2021) | − | 94.36 ± 0.11 | 93.13 ± 0.14 | 92.02 ± 0.19 | 80.82 ± 0.15 |

## 4.3 EXPERT BREAKDOWN

Our next goal is to understand the effect of different concepts on explaining the representation, *i.e.* answering the question: to which degree does the representation *know* certain concepts, *e.g.*, material? As we focus on self-supervised representations, the answer to this question is far from obvious. We measure this via the predictive ability of our probe, which we now train individually for each group of concepts. These vary in nature; from highly semantic object categories (*e.g.*, dog, avocado) to lower-level features such as material (*e.g.*, wood, brick), texture (*e.g.*, bubbly, striped) and color (*e.g.*, red).

In Fig. 4, we show how the different concept groups contribute to the overall performance for selected ResNet-50-based methods. We train and evaluate reverse probes for each shown combination, *e.g.*, IN-1K+OBJECTS, then IN-1K+OBJECTS+SCENE, etc. We compare variants with and without using ground truth IN-1K categories as part of the input. We observe that for earlier methods, such as MoCo-v1, ImageNet categories alone are not sufficient for accurately predicting the cluster assignments. In fact, it appears that using only semantic categories from MSCOCO and OpenImages (w/o ImageNet) provides about as much information about the clusters. This likely suggests that the discovered clusters do not reflect fine-grained distinctions, since the most notable difference between IN-1K and the other semantic experts is the granularity of some categories (*e.g.*, dog breeds). On the other hand, the best performing methods do owe a big part of their performance to IN-1K concepts, meaning that they are able to recover clusters that are closer to the original label distribu-

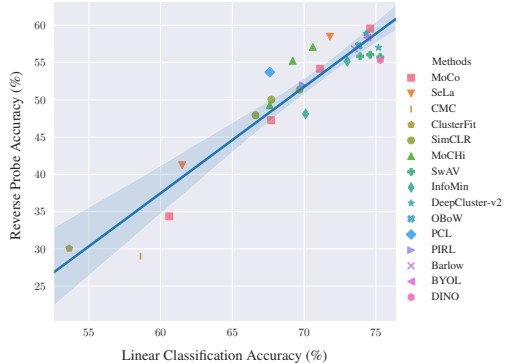 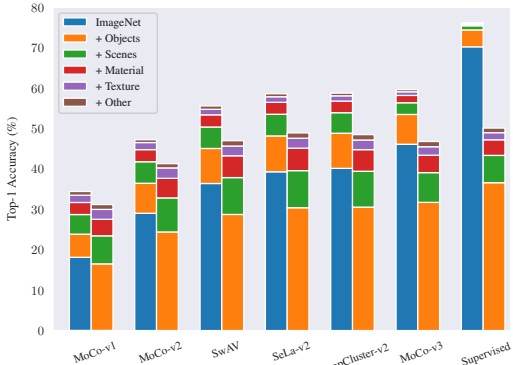

Figure 3: Linear classification accuracy on IN-1k vs. classification accuracy of our probe (top-1). A linear regression model is fit to the data, suggesting strong correlation. Each point corresponds to a ResNet-50-based model in Table 2.

Figure 4: Contribution of each expert group to the overall accuracy for selected methods. For each bar a linear model is trained using as input the current expert group and all previous ones (e.g., green bar: IN-1K+OBJECTS+ SCENES).

Table 3: Change in NMI when learning the reverse probe with individual concept groups on top of the human-annotated IN-1K labels.

| Method | | IN-1K | OBJECTS | SCENE | MATERIAL | TEXTURE | OTHER |
|---|---|---|---|---|---|---|---|
| MoCo-v1 | (He et al., 2020) | 47.70 | +2.73 | +2.92 | +2.93 | +1.04 | +0.21 |
| MoCo-v2 | (Chen et al., 2020b) | 57.39 | +2.43 | +2.09 | +2.03 | +0.43 | -0.24 |
| SwAV | (Caron et al., 2020) | 61.94 | +2.42 | +1.92 | +2.01 | +0.23 | -0.20 |
| SeLa-v2 | (Asano et al., 2020) | 63.29 | +2.23 | +1.69 | +1.69 | +0.03 | -0.45 |
| DeepCluster-v2 | (Caron et al., 2020) | 64.29 | +2.17 | +1.54 | +1.70 | +0.12 | -0.35 |
| OBoW | (Gidaris et al., 2021) | 64.73 | +2.77 | +1.82 | +2.13 | +0.11 | -0.15 |
| MoCo-v3 | (Chen et al., 2021) | 67.69 | +1.51 | +0.32 | +0.21 | -0.36 | -0.38 |

tion and that rely less on low-level concepts such as textures. This suggests that more performant methods learn features highly correlated with IN-1K labels, but less with other semantic categories.

Finally, in Table 3, we also show the (decorrelated) effect of each individual concept group *in isolation* but always in combination with the ground truth IN-1K categories. We observe that in absence of concept combinations, the models we probed use as much (and sometimes more) information about scene and material properties as information about semantic object categories. Miscellaneous concepts (OTHER) do not appear relevant for more recent methods. We again observe that the representations of the most recent, state-of-the-art MoCo-v3 share very little information with concepts other than IN-1K, despite the lack of label information during pre-training.

## 4.4 QUALITATIVE RESULTS

Since the clusters we examine are learned without supervision and expert models are not perfect predictors, it is crucial to verify that the clusters align with human interpretation, besides quantitative scores. By inspecting the probe's coefficients $\theta$, one can qualitatively explore what concepts are representative for specific clusters and how exposing the linear model to different sets of experts affects its predictive accuracy. In Fig. 5 we show pairs of clusters from MoCo-v2 for which we observed a significant drop in pairwise confusion (estimated from the confusion matrix) with the inclusion of an additional concept group on top of the fixed ImageNet label set. For example, after including OBJECTS we notice that several pairs of clusters become more easily disambiguated thanks to the concept `person`, or related concepts (`human hand`, `human face`, etc.). In Fig. 5 (top left), both clusters can be described by the ImageNet label `French horn` and it is only possible to explain their differences when more information becomes available. This finding supports our initial hypothesis that there could be interpretable properties in the network that remain undetected since they are not annotated *a-priori* in the benchmark data. Importantly, since ImageNet is only annotated with a single label per image, combined concepts, such as "person playing the french horn" cannot be discovered otherwise. Similarly, when adding SCENE concepts (top right), clusters

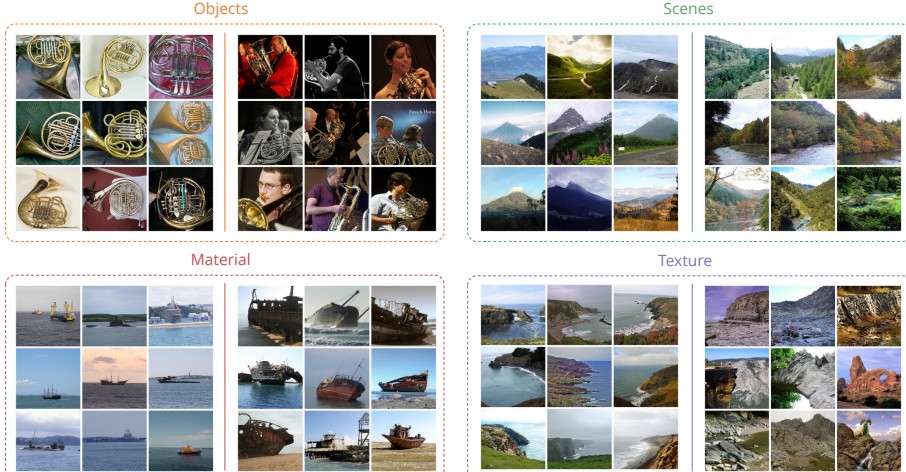

Figure 5: Qualitative examples. We show pairs of clusters for which the estimated confusion was significantly reduced after training the linear model with the corresponding concepts. We show and discuss additional examples in Appendix E.

Table 4: Evaluation of selected IN-1k-pretrained methods on the Places-365 dataset. $K$-means ($K = \{500, 1000\}$) and the reverse probe are trained on features extracted on Places-365.

| | $K = 500$ | | | | $K = 1000$ | | | |
|---|---|---|---|---|---|---|---|---|
| **Model** | **NMI** | **AMI** | **Top-1** | **mAP** | **NMI** | **AMI** | **Top-1** | **mAP** |
| MoCo-v2 | 54.41 | 51.84 | 41.18 | 42.15 | 54.81 | 47.34 | 34.05 | 32.74 |
| SeLa-v2 | 57.02 | 54.66 | 45.04 | 46.42 | 56.72 | 49.76 | 37.46 | 36.48 |
| SwAV | 57.30 | 54.65 | 44.84 | 46.48 | 57.96 | 50.44 | 39.08 | 38.34 |
| DeepCluster-v2 | **58.09** | **55.62** | **46.23** | **48.06** | **58.30** | **50.94** | **39.54** | **39.35** |

are distinguished by scene type; one contains `volcanos`, while the other contain `mountains` and `creeks` (categories from Places-365). With MATERIAL, we can predict the shipwreck cluster more accurately, while TEXTURE can explain the difference in the more `stratified` appearance of cliffs for the cluster on the right. Thus, it appears that, despite the object-centric nature of ImageNet, self-supervised representations rely significantly on *context* (*e.g.*, SCENE and MATERIAL), and concept combinations. This likely also explains why in (Van Horn et al., 2021) self-supervised models perform equally or better than the supervised baseline at context, counting and gestalt tasks, and why structural downstream tasks benefit more from self-supervision Kotar et al. (2021).

## 4.5 TRANSFER TO OTHER DATASETS

We finally evaluate the representations learned by selected methods by transferring them to a different domain. For this experiment, the self-supervised methods are *not* fine-tuned or adapted on the new data. We perform the quantization on representations extracted on images from Places-365 and use all concepts from Table 5 but IN-1K (now using the available ground truth for Places-365 instead). We report the performance in Table 4 and observe that the same ranking generally holds.

## 5 CONCLUSION

We have introduced reverse linear probing, a measure of representation interpretability that allows to explain representations by combining multiple concepts. Driven by the need to better understand and characterize the representations learned by self-supervised methods, the proposed measure has a rigorous foundation in information theory and provides complementary information to existing benchmarks as well as insights into the semanticity and the importance of different concepts for such models. Our approach is applicable to any representation and is considerably faster to train, as all concepts can be pre-extracted on any given image collection.

## ACKNOWLEDGEMENTS

I.L. is supported by the European Research Council (ERC) grant IDIU-638009 and EPSRC VisualAI EP/T028572/1. Y.M.A. is thankful for funding from EPSRC Centre for Doctoral Training in Autonomous Intelligent Machines & Systems (EP/L015897/1) and MLRA from AWS. A.V. is supported by the ERC grant IDIU-638009.

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

## A  Experts: Implementation Details

We provide all information about the concepts and experts used in our experiments in full detail. Expert models are trained on various tasks and image datasets to extract information about the scene and its objects, material, texture and miscellaneous (other). Then, the trained models are applied to the target dataset (*e.g.*, ImageNet (Russakovsky et al., 2015)). While the tasks differ in nature — *e.g.*, segmentation (dense), detection, classification — in all cases we convert the predictions to binary vectors. For example, in the case of segmentation and detection, we return only a (binary) label vector denoting whether a class is present in a given image, discarding dense/location information. We concatenate the experts' binary predictions forming the concept vector $y(x)$ for image $x$, *i.e.*

$$y(x) = [y^{(1)}(x); y^{(2)}(x); \ldots; y^{(m)}(x)],$$

for $m$ expert models with $y^{(i)}$ denoting the $i$-th expert. In the following, we provide some expert-specific details and summarize them in Table 5.

**Detection on Open Images.**  We use the Tensorflow Object Detection API and publicly available Faster R-CNN model (Ren et al., 2016), trained on Open Images (600 object categories). We apply the model to our target datasets and keep only object categories predicted with a confidence higher than 0.5.

**Segmentation on MS COCO.**  We use a DeepLab-v2 model (Chen et al., 2017) trained for segmentation on MS COCO 2017 (Lin et al., 2014; Caesar et al., 2018), which includes 91 'thing' and 91 'stuff' categories. 'Thing' categories include common, countable objects with a well-defined shape, such as person, dog, bicycle, etc.; 'stuff' categories typically include background regions, such as sky, snow, wall, grass, etc. and are thus closely related to material properties. While we experimented with confidence and minimum area thresholds for a predicted class to be considered in our scenario, we found that simply returning all predicted categories performed best.

**Scene Classification and Scene Attributes.**  Although ImageNet is an object-centric dataset, a lot of images are related to specific scene types, *e.g.*, outdoor scenery (mountains, seaside) or indoor scenery (houses, shops). In fact, some ImageNet categories can even further subdivided by scene type, for example dogs sitting *indoors* or playing in the *park*. It is thus crucial to consider scene classification in our investigations. We use a pre-trained ResNet-50 model provided by the official repository of the Places-365 database (Zhou et al., 2017) and assign each image of our target dataset to a scene category. Further, we use the provided unified model (wide ResNet-18) to predict multiple scene attributes per image. The model is trained on the SUN Attribute Database (Patterson and Hays, 2012), which contains 102 categories describing scene properties, *e.g.*, man-made vs. natural or enclosed vs. open area.

**Texture and Material Classification.**  For further detecting material concepts, we train a Deep Encoding Network (DEP) (Xue et al., 2020) on the Materials in Context (MINC) dataset (Bell et al., 2015), which consists of 23 categories (include a background class). We then do the same on the Describable Textures Dataset (DTD) (Cimpoi et al., 2014), which consists of close-up images for 47 texture categories. DEP uses ResNet-50 (He et al., 2016) as backbone and is trained for single-label classification. We apply the trained models to our target datasets, thus returning a single label per image in each case.

**Text Detection.**  Next, we wish to identify the presence of text in the images to investigate whether this is a deciding factor in clustering the image features (*e.g.*, camera dates or watermarks). As a text detection expert, we use CRAFT (Baek et al., 2019) (official implementation) without the refinement step. We only return one bit representing whether the image contains text or not, rather than character recognition. Overall, we found that text as a concept did not contribute significantly in our experiments.

**Sentiment Analysis.**  We use a VGG-19 classifier (Simonyan and Zisserman, 2014) trained on the Twitter for Sentiment Analysis Dataset (T4SA) to classify the sentiment for each image into one of three classes (positive/neutral/negative) and use the sentiment prediction as an additional concept. As with text, we found that sentiment does not significantly affect the performance. We found the predictions of the expert model to be somewhat unreliable; this is also likely due to the fact that sentiment is a rather subjective quality.

Table 5: Summarization of expert models and datasets used in our work for concept extraction.

| Category | Datasets | #Classes | Task | Model | Source |
|---|---|---|---|---|---|
| OBJECTS | Open Images | 600 | Segmentation | Faster R-CNN | * |
| | MS COCO (thing) | 91 | Detection | DeepLab-v2 (ResNet-101) | * |
| SCENES | Places-365 | 365 | Single-label Classification | ResNet-50 | * |
| | SUN Attribute | 102 | Multi-label Classification | WideResNet-18 | |
| MATERIAL | MS COCO (stuff) | 91 | Segmentation | DeepLab-v2 (ResNet-101) | * |
| | MINC | 23 | Single-label Classification | DEP (ResNet50) | * |
| TEXTURE | DTD | 47 | Single-label Classification | DEP (ResNet-50) | * |
| | Color | 11 | Quantization | - | * |
| OTHER | SynthText, IC13, IC17 | 1 | Text Detection | CRAFT | * |
| | T4SA | 3 | Sentiment Classification | VGG-19 | * |
| | OpenAI Data (unreleased) | 9 | Image/Text Similarity | CLIP | * |

**Photography.** We have observed differences in the photographic style, quality and type of images in ImageNet, *e.g.*, ranging from macro photography, usually for flowers or insects, to vector illustrations. In order to probe the representations for style and image quality, we use the recent CLIP model (Radford et al., 2021), trained on 400 million image-text pairs collected from the web and create a set of candidate sentences: "a high quality photo", "a noisy, grainy image", "a blurry image of low quality", "macro photography", "a photo with out of focus background, bokeh effect", "an animated picture, a vector illustration", "a painting", "a portrait". We use the pre-trained CLIP model to compute image and sentence features (normalized to unit norm) and use the dot product to find the similarity between each image and each one of the sentences. We pick the sentence with the highest similarity as the predicted class for each image if the score is at least 0.5, else we assign a background class. Since CLIP operates on free-form text rather than a fixed set of categories, its use is not limited to this type of queries; it can be used as an expert where a labelled dataset might unavailable.

## B  SSL MODELS

We have evaluated a wide array of self-supervised models, which we divide into the following categories.

(⊕⊖) **Contrastive methods** make use of positive and negative examples and learn representations by drawing positive samples together while pushing negative samples apart. Based on the idea of instance discrimination, positive examples are constructed from different views of the same image, thus the representation learns invariance to the choice of transformations, which are typically rather aggressive. We have evaluated MoCo, SimCLR, CMC, InfoMin and MoChi as contrastive approaches.

(⊕) **Positive-only methods** do not discriminate between instances, eliminating the need for negative examples altogether. However, simply removing negative examples can result in collapse (features can be a single constant vector). For this reason methods that use only positive examples avoid feature collapse through other techniques such as regularization or distillation. From this family of methods, we have evaluated BYOL, DINO and Barlow Twins.

(⊕) Another family of methods can be categorized as **clustering-based**. Clustering is another way to enforce invariance and it relies on the assumption that meaningful groups exist in the data, such that intra-group similarities can be maximized and inter-group similarities should be low. Offline clustering methods typically alternate two steps, *i.e.* assign datapoints to clusters based on their representations and optimize the model given the current cluster assignments. Online approaches, like SwAV, perform clustering in a minibatch and only enforce consistent assignments for different views of the same image. We have evaluated ClusterFit, SeLa, DeepCluster-v2 and SwAV as clustering-based approaches, as well as PCL which combines clustering and a contrastive objective.

(♣) Earlier methods on self-supervised visual representation learning devised **handcrafted pretext tasks**. As a representative of this category, we have evaluated Jigsaw, which trains the model to

solve puzzles as the pretext task. We also evaluated PIRL, which combines the Jigsaw task with NCE, to learn representations that are invariant to the input perturbations.

## C  TRAINING DETAILS

**Linear model**   We evaluate representations from a number of pre-trained self-supervised feature extractors (Table 2). For ResNet-50 models, we evaluate features at the output of the average pooling layer (2048-d). For ViT-Base models we evaluate the `[CLS]` token of the last self-attention layer (768-d). For each model, we pre-extract and store features for the entire training set of ImageNet and, prior to clustering, we standardize them to zero mean and unit standard deviation. We then run $K$-means for 100 steps using `faiss` (Johnson et al., 2017) and choose the best out of 5 runs. We divide the data into train and test sets by splitting all cluster assignments with a 80/20 ratio and stratified sampling; from the training split we also reserve 20% of the data for validation. Finally, we train the linear model with the cluster assignments as targets and concept vectors as inputs (where the concepts are pre-extracted on a given image collection, *e.g.*, ImageNet). We train for up to 100 epochs with batch size 512 and optimize using SGD with a momentum of 0.9 and initial learning rate of 3.5 which is further reduced by a factor of 10 at epochs 60 and 80. We also add L2-regularization with weight $3 \times 10^{-6}$.

**Computation**   Given a dataset (*e.g.*, ImageNet) and a set of experts (*e.g.*, the ones listed in Table 5), all labels can be pre-computed for all images in the dataset. Feature vectors for a given model can be also pre-extracted and stored for the whole dataset. Our method computes cluster assignments using the efficient $K$-means implementation of `faiss` (which takes less than 5min for 256k 2048-d vectors on 4 NVIDIA RTX A4000). Training of the linear model converges in less than 100 epochs in a matter of minutes on a single GPU (1 epoch takes 2sec). We should note that standard linear probing on ImageNet typically requires full images (including online augmentations) and multiple forward passes through the frozen feature extractor, which makes it significantly slower to train.

## D  FURTHER ANALYSIS

In addition to the results and discussion in the main paper, we provide more investigations and insights.

### D.1  MUTUAL INFORMATION BETWEEN CONCEPTS AND IMAGENET CATEGORIES

One interesting question that arises is to which extent the elementary concepts we are considering are predictive of ImageNet labels. We answer this question by training a linear model to predict the ground truth label (instead of a pseudo-label) for each image from its concepts. This results in top-1 accuracy of 46.8% (NMI: 64.2, AMI: 54.2) — and significantly less if we exclude 'object' concepts that overlap with ImageNet labels. This suggests that using additional concepts provides information which is *complementary* to the fixed label set of ImageNet, further justifying our approach.

### D.2  VARYING $K$

As discussed in Section 3, the mutual information between a representation $f(x)$ and a fixed set of concepts $y(x)$ is maximized when $f(x) = x$, *i.e.* any processing of $x$ cannot increase the amount of information. It is thus expected that with an increasing number of clusters $(K)$, $I(f_K(x), y(x))$ will monotonically increase, as we approximate $f(x)$ (*i.e.* when each sample is its own cluster). We verify this empirically in Fig. 6. For a fair comparison across methods, we have fixed $K = 1000$ for all experiments on ImageNet experiments in the main paper. To better understand the effect of the number of clusters, in Fig. 7 we also show the performance of most methods for $K \in \{500, 1000, 1500, 2000, 2500, 3000\}$, measuring NMI, AMI and top-1 accuracy, for the predictions of the reverse linear probe trained with all concepts. Importantly, we observe that for $K \geq 1000$ the relative ranking of methods remains mostly *consistent* regardless of the number of clusters. The top three methods, MoCo-v3, OBoW and DeepCluster-v2 perform similarly and converge for larger $K$, followed by SeLa-v2, SwAV, BarlowTwins and DINO. Further, we observe that BYOL scales gracefully for larger $K$, reaching the performance of the top methods; the opposite is observed for

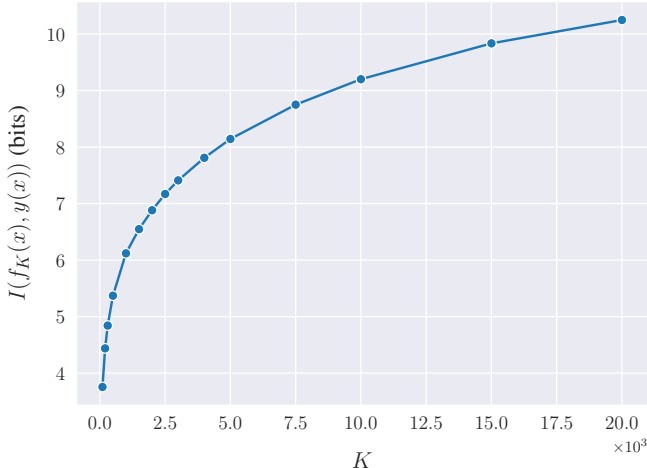

Figure 6: Empirically estimated mutual information between $f_K(x)$ and $y(x)$ for varying $K$.

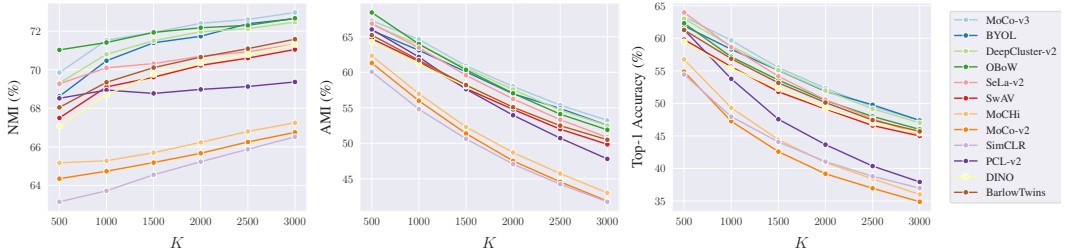

Figure 7: Performance of self-supervised methods (architecture: ResNet-50) for varying number of clusters $K$. The ranking remains mostly consistent.

PCL-v2 which does not scale as well. Though it is important to study the effect of varying $K$, we should also note that our results do not appear sensitive to it, as similar trends are observed for all methods.

### D.3 DISCUSSION ON CLUSTERING-BASED METHODS

We have observed that methods that employ a clustering mechanism during training have generally more interpretable representations. A natural question that arises is whether clustering-based evaluation naturally favors such approaches.

However, clustering has already been shown to work well even for contrastive approaches (Zheltonozhskii et al., 2020), which is also further confirmed by our experiments. In our evaluations, the highest ranked models are MoCo-v3 (1k epochs) and OBoW (200 epochs). MoCo-v3 uses a contrastive objective, while OBoW follows a different approach from previous methods with a bag-of-words prediction task that puts emphasis on contextual information and local feature statistics. Our results align with the intuition behind OBoW (Gidaris et al., 2021) that encoding local concepts via a bag-of-visual-words approach yields richer, context-aware representations. Another example, SeLa, uses Sinkhorn-Knopp optimization to compute the cluster assignments, while DeepCluster-v2 uses spherical K-means. All these clustering mechanisms differ from the one used specifically in our experiments. Moreover, SeLa and DeepCluster-v2 train with $K = 3000$ prototypes, and there is no reason to believe that their representations are optimal for all values of $K$, and as we discussed above, the relative ranking of methods does not depend on the choice of $K$.

| Method | | NMI | AMI | Top-1 | mAP |
|---|---|---|---|---|---|
| SeLa-v1 ($K$-means) | (Asano et al., 2020) | 64.66 | 37.37 | 33.05 | 29.29 |
| SeLa-v1 (self-label) | (Asano et al., 2020) | 62.02 | 31.92 | 29.22 | 24.51 |
| SCAN ($K$-means) | (Van Gansbeke et al., 2020) | 69.02 | 61.56 | 54.17 | 60.44 |
| SCAN (self-label) | (Van Gansbeke et al., 2020) | 73.76 | 73.76 | 62.10 | 63.63 |

Table 6: Evaluation of self-labelling methods. "Self-label" denotes clusters which are predicted *directly from the model* and which are used to train the reverse probe. We compare these to the clusters obtained with $K$-means on the representation vectors (as in the main paper). $K = 3000$ for SeLa and $K = 1000$ for SCAN to match the number of pseudo-labels predicted by the respective models.

| Method | Top-1 | Top-5 | NMI | AMI |
|---|---|---|---|---|
| MoCo(v2) | 40.30 | 75.57 | 60.14 | 50.38 |
| | 12.67 | 32.45 | 26.73 | 16.53 |
| MoCHi | 41.03 | 75.62 | 59.89 | 50.48 |
| | 12.64 | 31.71 | 25.66 | 15.98 |
| SwAV | 45.73 | 78.56 | 62.75 | 53.77 |
| | 17.71 | 43.64 | 26.39 | 18.72 |
| OBoW | 46.92 | 78.73 | 64.82 | 55.75 |
| | 12.23 | 31.72 | 27.45 | 16.82 |
| DeepCluster(v2) | 47.20 | 79.90 | 63.25 | 54.77 |
| | 19.73 | 46.81 | 26.84 | 19.43 |
| SeLa(v2) | 47.62 | 80.87 | 62.88 | 54.91 |
| | 20.61 | 48.83 | 25.32 | 18.36 |

Table 7: Reverse probes evaluated on ObjectNet in comparison to ImageNet (in black).

## D.4  $K$-MEANS FOR SELF-LABELLING METHODS

We now look specifically at self-labelling methods, SeLa (Asano et al., 2020) and SCAN (Van Gansbeke et al., 2020), to investigate the difference between (a) the pseudo-labels predicted directly by each method and (b) clustering the learned representations with $K$-means, *i.e.* same as the experiments reported in the main paper. SeLa follows an optimal transport approach that yields clusters of approximately equal size. We found that this weakens the quality of its self-labels but results in stronger clusters after $K$-means and consequently in improved performance in our evaluations. Since SeLa is originally trained with 3000 classes[1], in Table 6 we compare its pseudo-labels with $K = 3000$-means clustering in Table 6. SCAN builds on top of a representation learning method, *i.e.* MoCo (He et al., 2020) and, with a learnable clustering approach that removes adverse effects of low-level features, it produces more semantic pseudo-labels, improving upon $K$-means. Thus SCAN (self-label) results in improved performance in Table 6.

## D.5  TESTING ON OBJECTNET

Next we provide an evaluation of selected self-supervised methods using the reversed probes on the challenging test set ObjectNet (Barbu et al., 2019), evaluating the transferability of the quantized representations. Specifically, we use the the reversed probes trained for each method on ImageNet to assign ObjectNet samples to clusters, measuring the success of the classification (accuracy, NMI, AMI) against the corresponding $K$-means assignments, *i.e.* we measure if we can successfully predict the cluster assignments from predicted concepts, without any further training/adaptation. Here, we use the probes trained with all experts as input, except for the real ImageNet labels to account for

---

[1] https://github.com/yukimasano/self-label

ObjectNet categories that do not appear in ImageNet. We notice a significant drop in performance which suggests that a domain gap is indeed present, though the ranking stays again roughly the same (with the exception of OBoW).

# E  QUALITATIVE EXAMPLES

Finally, in Figs. 8, 9, 10 and 11 we present several additional qualitative examples that highlight the usefulness of reverse probing. Similar to Fig. 5 in the main paper, we begin by showing self-supervised clusters which are similar; they often contain images belonging to the same ImageNet categories. As a result, when training a (reverse) linear probe from ImageNet labels to pseudo-labels (cluster assignments), pairs of clusters with overlapping concepts will typically have high confusion (computed from the confusion matrix). We then add a concept group on top of the ImageNet labels and train a second linear probe, *e.g.*, in Fig. 9 we include concepts from Places-365 and SUN Attributes. We can then identify pairs of clusters for which the inclusion of these concepts significantly reduced the ambiguity of the mapping, therefore reducing the confusion. Finally, for each pair of clusters we show the difference in linear model's coefficients as a word cloud, with larger differences denoted by larger font size. In other words, we show the concepts that are most important to distinguish the two clusters (blue for (i) and red for (ii)).

In this way, we were able to discover and understand fine-grained differences between clusters and even problematic cases when considering only ImageNet labels. For example, we find that several ImageNet categories appear in combination with people (*e.g.*, musical instruments). The quantized space of self-supervised representations (for all methods) appears to separate the corresponding images based on whether they contain people or not (examples shown in Fig. 8 (a), (c) and (d)). Noteworthy are the clusters in Fig. 8 (d), both of which correspond to the same ImageNet label (`hockey puck`), although they are visually very different. Including additional concepts, such as `person` or `sports equipment`, justifies their distinction into separate clusters and aligns with human intuition — perhaps even more so than assigning them to the same class. In Fig. 9 we show examples where clusters with images from the same ImageNet category differ by scene-related concepts such as flying vs. perched birds, wolves and orcas in different environments and even the inside vs. outside of a building. Material categories (Fig. 10) are helpful to understand environment or context and whether images contain hands (`skin`). Finally, texture (Fig. 11) can be used to tease apart details at a macro level, *e.g.*, cut and whole fruits or peacocks with open or closed plumage. Overall, texture becomes less relevant models which are better at semantic discrimination. We also found that other concepts, such as text, sentiment and image quality do not significantly affect the clustering.

To summarize, we have proposed reverse linear probing as a way to understand whether interpretable clusters form in the representation space of self-supervised methods. While we provide a quantifiable measure for this — more interpretable clusters will be predicted with higher accuracy — we also show that we can qualitatively identify how concepts are encoded in the quantized representations through the linear probe's coefficients. We finally show that such concepts are often complementary to ImageNet labels and can in fact *correctly* push clusters which are seemingly semantically close — based on a fixed label set — farther apart (*e.g.*, Fig. 9(d)).

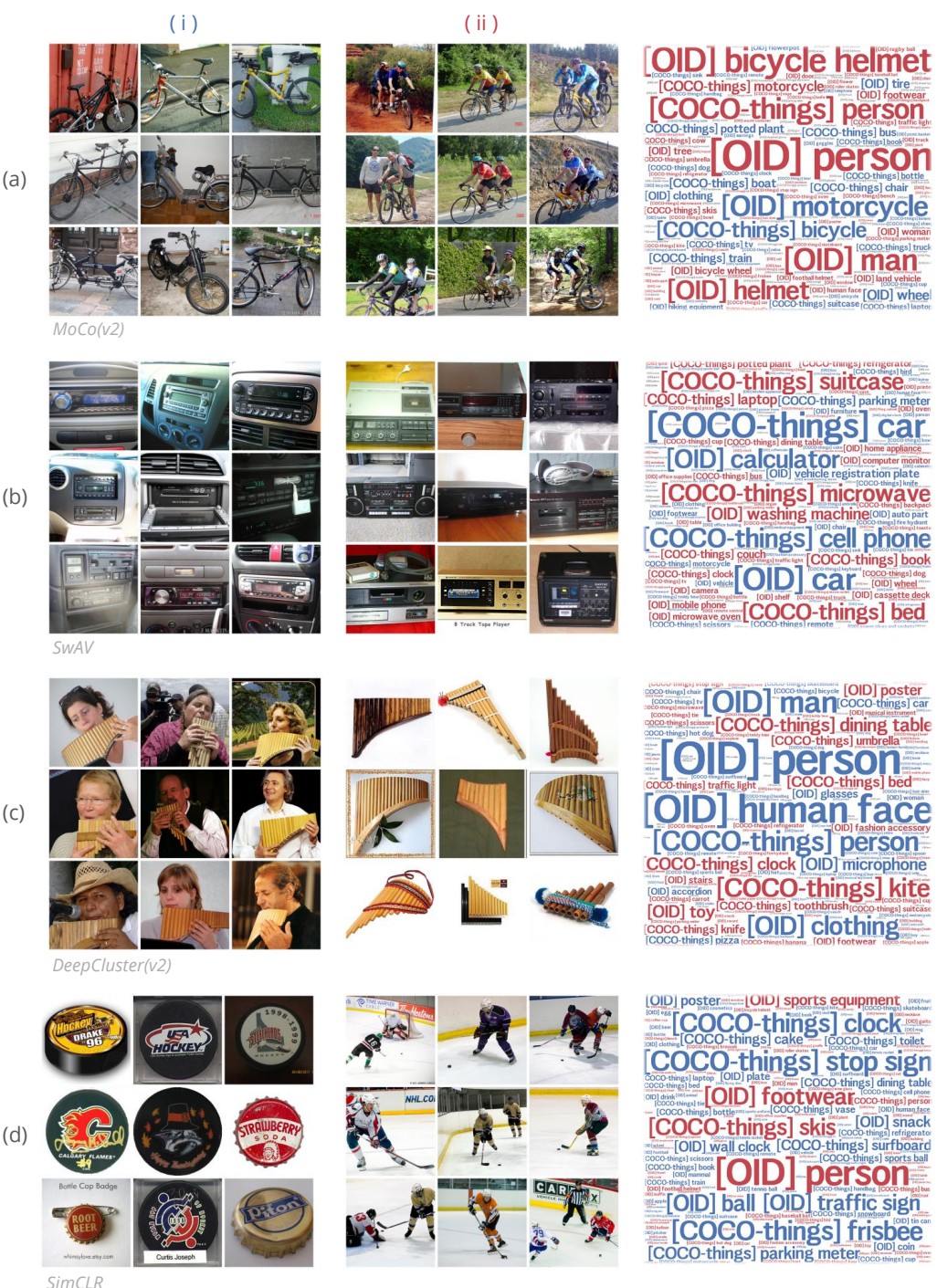

Figure 8: Clusters found by unsupervised methods, where each pair (i)-(ii) contains the same (or similar) ImageNet label(s). Confusion is high when probing with ImageNet categories alone, but is significantly reduced after including **Object** concepts from experts trained on COCO-thing and Open Images (OID) categories. The word clouds show the difference in the regressor coefficients for each pair (absolute value is denoted by increasing font size with blue: (i) > (ii) and red: (ii) > (i)).

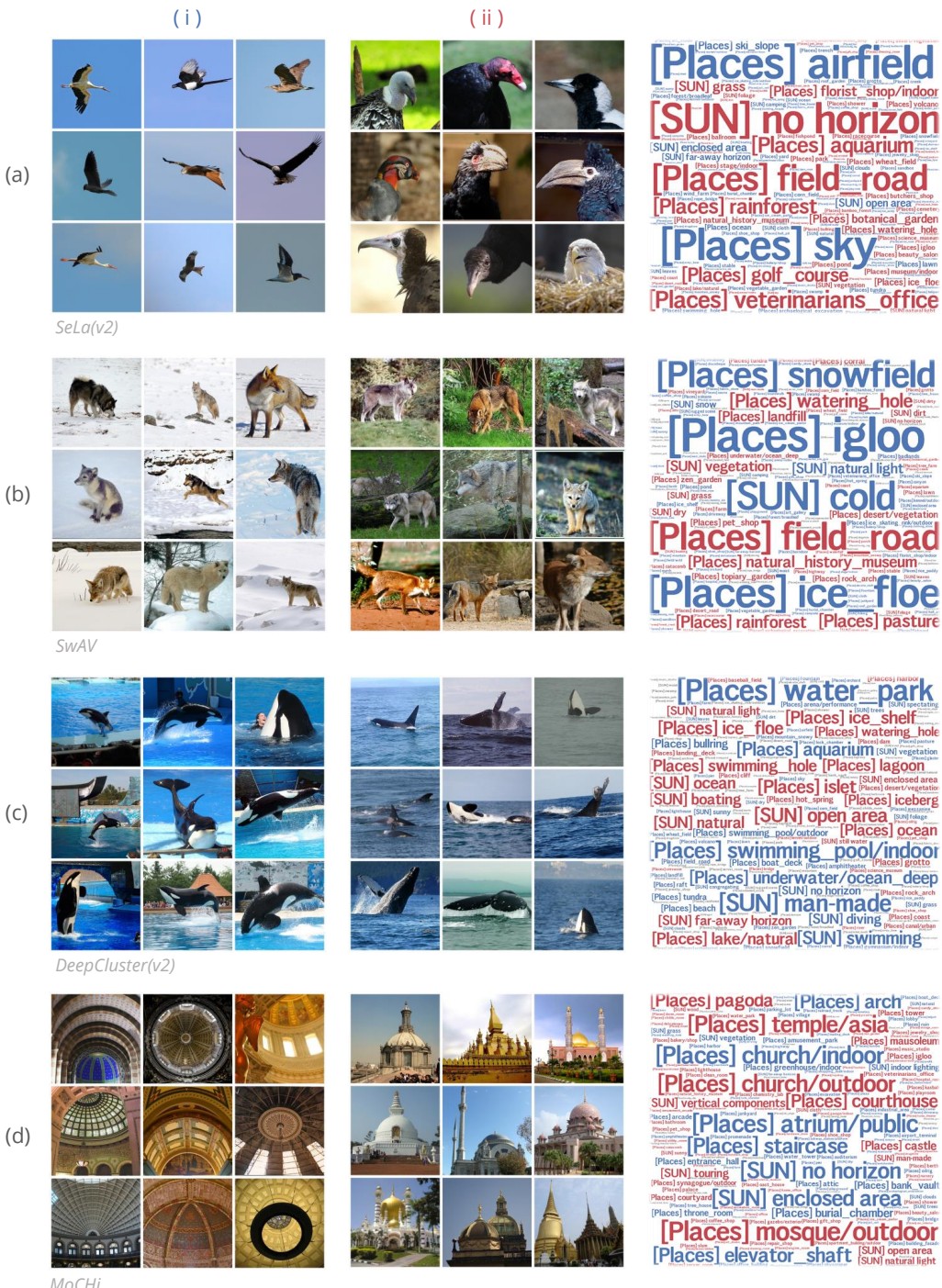

Figure 9: Clusters found by unsupervised methods, where each pair (i)-(ii) contains the same (or similar) ImageNet label(s). Confusion is high when probing with ImageNet categories alone, but is significantly reduced after including **Scene** concepts from experts trained on Places-365 categories and SUN Attributtes. The word clouds show the difference in the regressor coefficients for each pair (absolute value is denoted by increasing font size with blue: (i) > (ii) and red: (ii) > (i)).

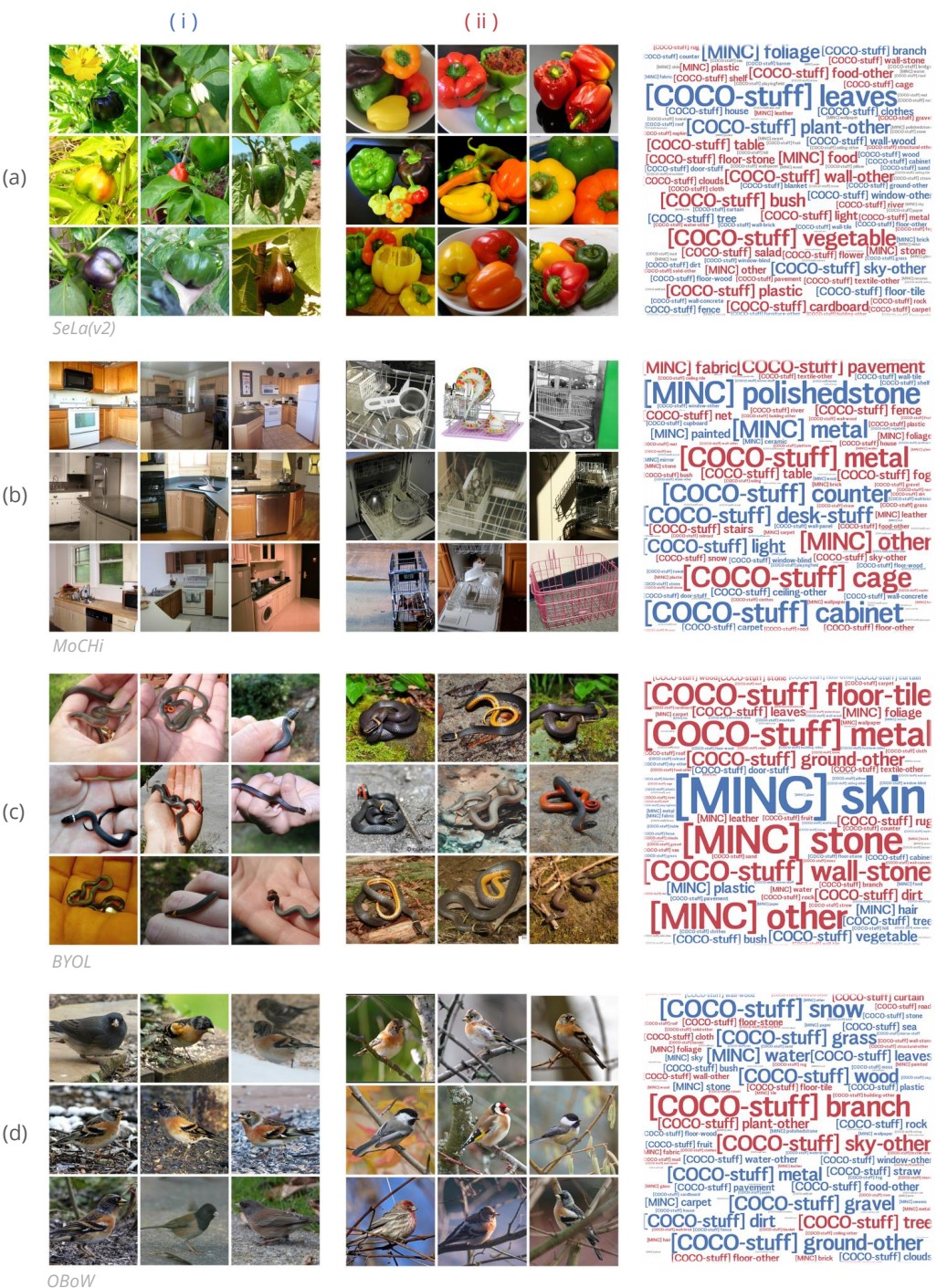

Figure 10: Clusters found by unsupervised methods, where each pair (i)-(ii) contains the same (or similar) ImageNet label(s). Confusion is high when probing with ImageNet categories alone, but is significantly reduced after including **Material** concepts from experts trained on COCO-stuff and Material in Context (MINC). The word clouds show the difference in the regressor coefficients for each pair (absolute value is denoted by increasing font size with blue: (i) > (ii) and red: (ii) > (i)).

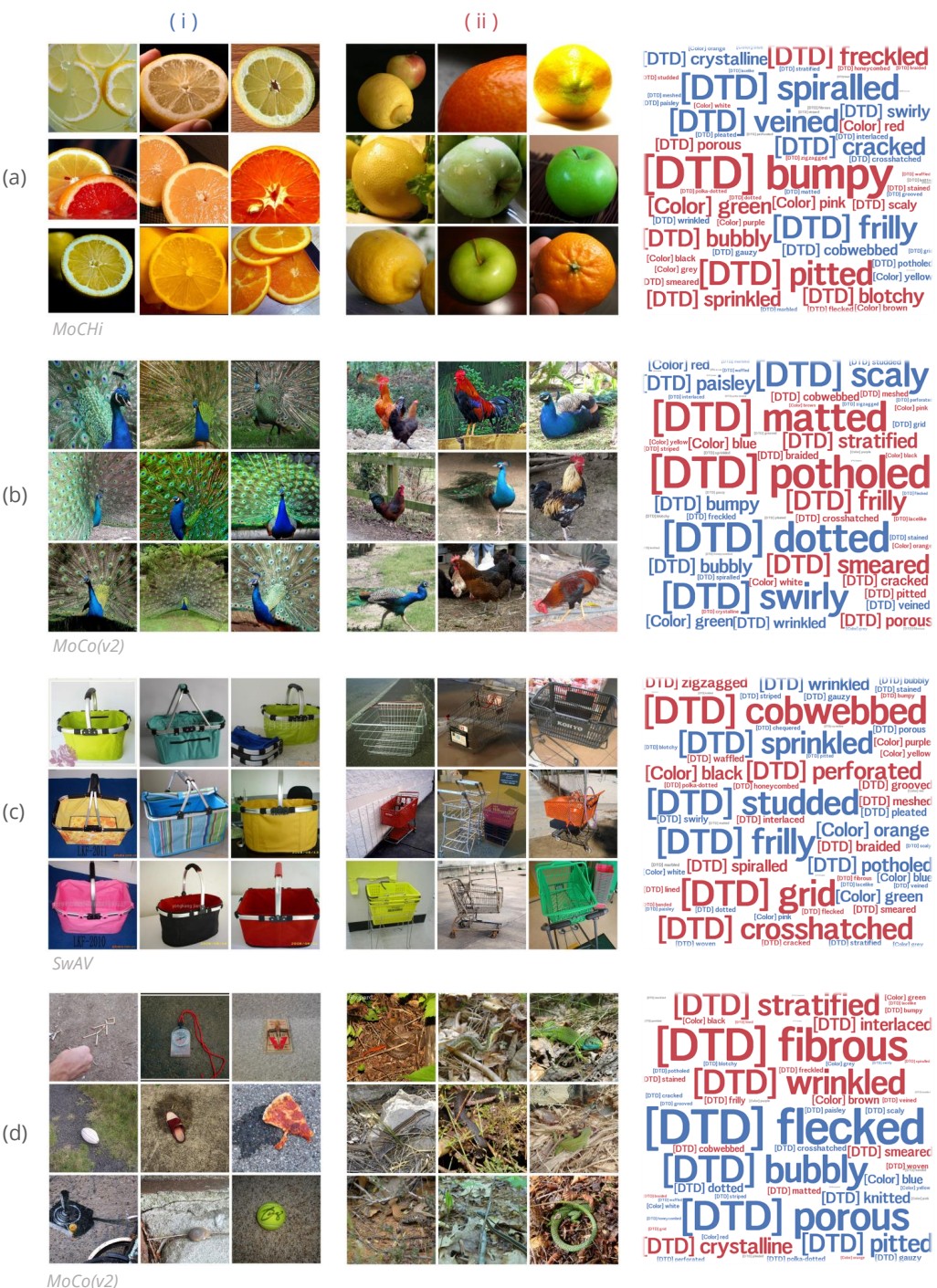

Figure 11: Clusters found by unsupervised methods, where each pair (i)-(ii) contains related ImageNet label(s). Using **Texture** concepts from experts on the Describable Textures Dataset (DTD) and 11 elementary colors helps reduce the confusion comparing to using only ImageNet categories. Clusters where texture helps are usually of lower quality and more recent methods depend less on textures.

## F    RELATION TO TOPIC MODELS

Probabilistic topic models, such as LDA (Blei et al., 2003), often used in natural language processing, aim to recover the latent semantic structure, *i.e.* a set of abstract "topics", from a collection of documents. The goal is to find the collection of the topics that best represent the corpus, with each document typically containing multiple topics in different proportions. However, the unsupervised nature of these models and the lack of a gold standard makes their evaluation a long-standing problem. For this reason, several automatic "topic coherence" measures have been proposed that evaluate the degree of semantic similarity of the top words in a topic (Mimno et al., 2011; Aletras and Stevenson, 2013; Newman et al., 2010; Röder et al., 2015), while human evaluations are also common (Chang et al., 2009; Morstatter and Liu, 2018).

There is a similarity between topic coherence and the way we evaluate models. However, this analogy breaks down rather quickly. To discuss this, we can directly map the terms of topic models (topics, documents and words) to our scenario (clusters, images and concepts) in this order. The semantic coherence of a topic (cluster) is often measured by the co-occurrence of words (concepts) across documents (images). One immediate difference that arises is that in topic modeling the original space (corpus) is authored by humans and can be thus considered interpretable. Measures of coherence are introduced to understand whether the topics found by a model are also interpretable, *i.e.* evaluating the model producing the topics. In our case, we do not know the degree to which the original space (representation) is interpretable and this is precisely what we aim to quantify with our method.

Another conceptual difference is that topic models operate on a higher level of abstraction: a topic is a collection of words that describe a higher-level idea defined by this collection of words. This is sensible as the goal in topic modelling is to discover which words describe higher-level ideas and can be likely grouped together vs. irrelevant words.

In our case, this is not necessary, as the expert annotations already define the relevant concepts. A further abstraction is unnecessary and in fact it may even be undesirable because we are interested in the minimum number of concepts that may explain a cluster. For example, a topic of "animals" may be coherent, yet a cluster of "animals" lacks specificity because it may contain a multitude of classes.

Finally, the main mechanism in evaluating topic coherence is co-occurrence of words in the general corpus (documents). In our case, this translates to visual concepts co-occurring in the same image and models relationships between concepts in the images/the real world, whereas we are interested in the relationship between clusters and concepts as they have been learned by the model.

