# OpenReview forum: "Measuring the Interpretability of Unsupervised Representations via Quantized Reversed Probing"
_ICLR.cc/2022/Conference — ICLR 2022 Poster_

### Official Review · Reviewer_AZhK · 2021-11-02

**Correctness:** 3
**Technical Novelty And Significance:** 3
**Empirical Novelty And Significance:** 3
**Recommendation:** 6
**Confidence:** 4

**Details Of Ethics Concerns:**

The proposed idea is sound and, to the best of my knowledge, novel.
I believe the proposed method can perfectly complement the somewhat reduced amount of efforts along the line of interpretation of pre-trained models. However, I believe several aspects (please see my review) that should be polished/clarified before the paper is ready for publication.

**Main Review:**

On the stronger side, the proposed method aims at addressing a task, post-hoc interpretation of pre-trained models, that had received much reduced attention compared to its explanation counterpart. As such it could help push research in this direction further.
In addition, the proposed method is evaluated considering a good amount of self-supervised learning methods.
Finally, code related to the paper will be released upon the publication of the manuscript which is always from the point of view of reproducibility.

On the weaker side,

While the idea of having the proposed probing in reverse order is, to best of my knowledge novel to me, it is not clear to what extent the proposed method does provides insight in the representation learned by the model being probed. Is is by providing semantic means (via y(x)) to describe some modes (f_K(x)) in the representation f(x) ? Perhaps it should be made more explicit the way in whic the proposed method makes the interpretation, of the model being analyzed, possible.

I find the way in which the paper approaches the idea of finding relationships between the internal representations learned by a model and a set of concepts very similar to [Escorcia et al., 2015], that did it at the attribute level, and to [Oramas et al., 2019], that further explore it towards interpretation of learned representations.
Unfortunately, there is no descriptive nor quantitative comparison with respect to these methods. Given the similarity, I would suggest properly positioning against them.

At this point it is not clear how the quantized representation $f_K(x)$ is actually computed. For instance, is the internal representation f(x) taken at the layer level, fed to the clustering algorithm and the its result concatenated with that from other layers. Or perhaps you feed the concatenation of all the internal activations to the clustering components, or only use a specific [set] of layers, e.g. the last convolutional layers? Are there any normalization steps involved?

In Section 3, it is mentioned that "we can further and cheaply increase the coverage of the label space by predicting the labels y(x)
automatically via a battery of expert classifiers learned using full manual supervision." While this indeed may seem as a positive and cheap means to further get annotations, as properly noted by the paper, it indirectly requires the existence of the expert predictors and the annotated data on which they were trained. While applicable in the more general imageNet like testing set considered in the experiments, this requirement might not in place to more specific/niche applications, which will limit the applicability of this practice.

At the end of Section 4.2, by using the proposed method, it is observed and stated that methods that use a clustering mechanism during training have generally more interpretable representations. Doesn't this observation originates from the fact that the proposed method relies on an intermediate clustering step and as such favors methods that rely in similar clustering steps as well. If this indeed the case, perhaps there is some bias indirectly injected by the method that favors some types of methods over others.

In Section 4.4, it is observed that confusion between different clusters decreases when additional concept groups are added. This observation is used to further stress the fact that there could be interpretable properties in the network that may remain undetected since they are not annotated a-priori in the benchmark data and that this further motivates the need for reverse linear probing.
In this regard, this is a known limitation of standard linear probing methods whose interpretation capabilities are bounded by the set of pre-defined annotated concepts. In my opinion, this is also a limitation of the proposed method which, as shown in Section 4.4. does require additional pre-defined concepts in order to decrease the confusion of the identified clusters. Therefore I do not agree with the statement that the proposed reverse linear probing is a solution to the problem.

Finally, it might be good to provide some details on the linear models mentioned at the end of Section 4.1. Is this a single [fully-connected] layer neural network trained for several epochs or perhaps something more elaborated?

**Summary Of The Paper:**

The paper proposes a method for characterizing the "meaning" of the representation learned by a given model (interpretability). Towards this goal, it proposes reverse linear probing, a post-hoc method that aims at predicting a quantized version of the internal representation (as observed in specific examples) from semantic label.

Special emphasis is given in the capability of the proposed method on enabling the interpretation of models learned in a self-supervised manner.

PROs
+ Code will be released upon acceptance.
+ Evaluation covers a good variety of models.

CONs
- Missing related literature
- Unclear how interpretability is gained.

**Summary Of The Review:**

The proposed idea is sound and, to the best of my knowledge, novel. I believe the proposed method can perfectly complement the somewhat reduced amount of efforts along the line of interpretation of pre-trained models. However, I believe several aspects (please see my review) that should be polished/clarified before the paper is ready for publication.

---

> ### Author Response · Authors · 2021-11-21
> **Response to Reviewer AZhK [2/2]**
>
> > **It is observed that methods that use a clustering mechanism during training have generally more interpretable representations. Doesn't this observation originates from the fact that the proposed method relies on an intermediate clustering step and as such favors methods that rely in similar clustering steps as well.**
>
> This is not necessarily the case, as clustering seems to work well even for contrastive approaches; this is also noted by (Zheltonozhskii et al. 2020). As a matter of fact, the highest ranked models are MoCo-v3 (1k epochs) and OBoW (200 epochs). MoCo-v3 uses a contrastive objective, while OBoW follows a different approach from previous methods with a bag-of-words prediction task that puts emphasis on contextual information and local feature statistics. Another example, SeLa, uses Sinkhorn-Knopp optimization to compute the cluster assignments and DeepCluster-v2 uses spherical K-means. Moreover, SeLa and DeepCluster-v2 train with $K=3000$ prototypes, and there is no reason to believe that their representations are optimal for all values of $K$; yet we find that the relative ranking of methods does not depend on the choice of $K$ (Figure 7).
> In all cases the expectation is that representation space contains semantically meaningful parts and our approach evaluates the extent to which this is true for all methods. Under our metric, we found that clustering-based methods do generally better than contrastive ones with a similar linear classification accuracy, meaning that using some notion of grouping during self-supervised training likely leads to more interpretable representations.
>
> > **The observation [that confusion between different clusters decreases when additional concept groups are added] is used to stress the fact that there could be interpretable properties in the network that may remain undetected since they are not annotated a-priori in the benchmark data and that this further motivates the need for reverse linear probing. This is a known limitation of standard linear probing methods — this is also a limitation of the proposed method which, as shown in Section 4.4. does require additional pre-defined concepts in order to decrease the confusion of the identified clusters. Therefore I do not agree with the statement that the proposed reverse linear probing is a solution to the problem.**
>
> We would like to clarify that there are two crucial aspects here. The first one is the importance of expanding the effective label set, so that it is possible to perform analysis over concepts that are not annotated a-priori in the benchmark data, _for the sake of interpretation._ A characteristic example here is that several clusters contain (prominently) the category “person”, which is not an annotated class in ImageNet. However, without bringing this concept into existence in the first place, it is not possible to understand its role in the representation. Expanding the label set can be realized for reverse probing, _as well as_ for standard linear probing. For example, the Broden dataset (Bau et al., 2017) has been created with a similar goal in mind. We see this as a vital step to interpretation rather than a limitation.
>
> The second aspect is the specific advantage of reversing the probing direction, which we have illustrated with an example in Figure 2 of the paper. In our revised version, we have made the discussion of this figure more comprehensive to better explain why the concept-to-representation direction handles combinations of individual attributes more effectively. We also clarify that what motivates the need for reverse linear probing in Section 4.4 is the observation that clusters in the representation space often contain such attribute combinations and not the availability (or lack) of annotations.
>
>
> > **Provide some details on the linear models mentioned at the end of Section 4.1. Is this a single [fully-connected] layer neural network trained for several epochs or perhaps something more elaborated?**
>
> For the sake of interpretability, we use a single fully-connected linear layer in all experiments and the training details for this model are provided in the appendix.

---

> > ### Comment · Reviewer_AZhK · 2021-11-30
> > **Re: Response to Reviewer AZhK**
> >
> > Thanks for addressing my review.
> >
> > To a good extent the review has addressed my concerns.
> > Having said that, there are very valid concerns pointed out by reviewer phRe, that I would advise explicitly addressing in a revised version of the manuscript.
> >
> > In light of the above I am considering increasing my initial rating.

---

> > > ### Author Response · Authors · 2021-11-30
> > > **Re: Re: Response to Reviewer AZhK**
> > >
> > > Thank you for your response!
> > >
> > > We are glad that we have addressed your concerns and we would appreciate it if you raised your rating to reflect this.
> > >
> > > We have also addressed all points raised by reviewer phRe and added the relevant discussions in the revised manuscript that we have uploaded (e.g., we have included a lenghty discussion on topic models in the appendix, categorized methods under a taxonomy and evaluated more methods with varying K to verify that this choice does not affect the ranking).

---

> ### Author Response · Authors · 2021-11-21
> **Response to Reviewer AZhK [1/2]**
>
> We thank the reviewer for the valuable feedback.
>
> >  **It is not clear to what extent the proposed method provides insight in the representation learned by the model being probed. Is it by providing semantic means (via y(x)) to describe some modes (f_K(x)) in the representation f(x)?**
>
> This is exactly right. Measuring the mutual information between $f_K(x)$ and semantic labels $y(x)$ provides a quantitative figure that represents the overall semanticity of the representation space of a model. This makes it convenient to probe and compare a number of models. More than that, considering different groups of concepts in isolation (objects, scenes, materials and texture), tells us the extent to which concepts of different nature are “present” in the representation. For example, a key insight is that more performant models’ clusters are less likely to contain low level attributes (e.g. color and textures), which is a desirable property when the goal is to learn highly semantic representations.
>
> Last but not least, there are qualitative ways to obtain insights into a learned representation. By examining the linear model’s coefficients, one can find _which_ concepts are the most important for each cluster in the representation space, which can be used to answer questions such as: is it a purely semantic concept or does the cluster carry lower-level information (e.g. indicated by high weights on texture and material attributes)? Or is the cluster related to a combination of semantic concepts, e.g. people and wind instruments (Fig. 5)? We show a number of examples in the appendix (Fig. 8-11), with an emphasis on pairs of clusters that share the same ImageNet label(s), but are visibly different, suggesting that ImageNet labels alone are not enough to understand the representation, despite the fact that it was learned on ImageNet. The difference is better “explained” with the inclusion of additional concepts in the evaluation, which are shown as word clouds. However, it is important to note that due to the high number of clusters (typically K=1000), an exhaustive qualitative analysis is not possible, which makes the use of a quantitative metric necessary to “summarize” the findings.
>
> > **The way in which the paper approaches the idea of finding relationships between the internal representations learned by a model and a set of concepts is similar to [Escorcia et al., 2015], that did it at the attribute level, and to [Oramas et al., 2019], that further explore it towards interpretation of learned representations.**
>
> Thank you for the references! We are happy to discuss these in our revised related work section. In fact, [Escorcia et al., 2015] is a good example for the _forward probe_ formulation, i.e. studies whether intermediate representations are predictive of attributes. [Oramas et al., 2019]  identify features relevant for the prediction of a given class, which is likely not applicable to unsupervised models.
>
> > **It is not clear how the quantized representation fK(x)  is actually computed.**
>
> Rather than analyzing all intermediate layers, we focus on the penultimate layer representation of a model. On ResNet-50 models this is the output of the average pooling layer, resulting in a 2048-d vector. On ViT models we evaluate the [CLS] token of the last self-attention layer (768-d vector). Prior to clustering we standardize the features to zero mean and unit standard deviation. We will include these details in the revision.
>
> > **"We can further and cheaply increase the coverage of the label space by predicting the labels y(x) automatically via a battery of expert classifiers learned using full manual supervision." — indirectly requires the existence of the expert predictors and the annotated data on which they were trained. While applicable in the more general imageNet like testing set considered in the experiments, this requirement might not in place to more specific/niche applications, which will limit the applicability of this practice.**
>
> We agree that our approach relies on some form of annotated data used to train expert predictors with supervision. However, we still think this is more feasible than annotating all possible concepts on any given dataset. In addition, the concept base could be seamlessly expanded as new datasets and/or expert models become available in the future.
>
> ImageNet is still the most common choice for self-supervised pre-training and, due to the large number of pre-trained models available, our efforts were focused in this direction. As self-supervised learning reaches various specific domains, e.g. the medical domain, it is likely that specialized probing methods will be necessary. However, we believe that we need to develop suitable algorithms for understanding and evaluating models, _before_ we apply self-supervised learning to downstream tasks or other domains.

---

### Official Review · Reviewer_UJ5M · 2021-11-02

**Correctness:** 2
**Technical Novelty And Significance:** 1
**Empirical Novelty And Significance:** 1
**Recommendation:** 3
**Confidence:** 4

**Main Review:**

Strengths

[S1] The paper tries to answer a very important question: to what extent the self-supervised visual representations can be interpreted by humans. This question is important and interesting as these representations were learned without human interventions but showed to be very effective for many downstream recognition tasks. While there are many task-oriented evaluations, systematic analysis of the interpretability of the representations is missing.

[S2] Formulating the “interpretability” as the mutual information between the representations and the semantic concepts is technically sound.

[S3] The analysis covers a wide range of visual concepts and pretrained self-supervised models.

[S4] The paper is well written.

—------------------------------------------------------------------------------------------------------------------------

Weaknesses

The proposed Quantized Reverse Probing is claimed to be complementary to linear probes with further advantages. At this point, I don’t think this claim is supported by the method or the experiments. To me, the proposed method is too similar to linear probes that they share the same drawbacks.

[C1] “A limitation of this type of approaches is that one can only discover in the representation meanings that are represented in the available data annotations”. I’m afraid both linear probing and the proposed method have this problem, the proposed method relies on pseudo labels generated by the expert models that were trained with data annotations.

[C2] “The representation might be predictive of combinations of attributes, i.e. it might understand the concept of ‘red apple’ without necessarily understanding the individual concepts, e.g. ‘red’ and ‘apple’.” This statement is correct. Unfortunately, neither linear probing nor the proposed method can resolve this problem, as they both are linear functions, so score(‘red apple’) == score(‘red’) + score(‘apple’) . In order to recognize “red apple”, both methods should have high scores on “red” and “apple”. In other words, the claim that the proposed method can better handle the compositions of the concepts is incorrect from the methodology perspective.

[C3] “The proposed method provides a single principled figure characterizing the entire representation instead of a separate figure for each predicted attribute”. With the proposed linear mapping, “this principled figure” is obtained by simply averaging the edge weights between the concepts and the clusters. I don’t see why linear probing cannot obtain such a figure by averaging the per-category scores.

[C4] “our measure can be computed much faster than linear probes”. Please elaborate more on this, as at first glance I thought the proposed method is slower as it involves clustering.

[C5] From the experiments, it is still not obvious to me that the proposed method is better than linear probing. Some observations listed in the paper are as expected or already pointed out by previous literature: deepcluster-v2 performs better as the optimization objective of deepcluster-v2 is pretty similar to the optimization objective of the proposed metric; also the evaluated self-supervised methods have very similar performance so their rankings may be sensitive to the metrics; ViT + Self-supervised-learning performs better on KNN classifications (and clustering ) was already observed by DINO, etc.

—------------------------------------------------------------------------------------------------------------------------

Minor comments:

Page 2, “either only cluster (Yan et al., 2020a) or cluster and further train and …”



**Summary Of The Paper:**

The paper aims to measure the interpretability of the visual representations learned by the recent self-supervised models. In doing this, the paper formulates the “interpretability” as the mutual information between the feature clusters and a set of visual concepts that can be interpreted by humans. The mutual information is approximated by the proposed Quantized Reverse Probing, a linear function that maps the concept vectors to the feature clusters. The proposed method is claimed to be complementary to linear probes, with further advantages: (1) a single principled score rather than individual prediction scores for different labels in linear probes (2)  handle combinations of concepts better than linear probes. (3) Faster than linear probes.

**Summary Of The Review:**

While the problem studied in the paper is very interesting, at this moment, I don't see there are any obvious advantages of the proposed method over a simple linear probing, as discussed in the "weaknesses". Therefore, I'm leaning toward a rejection.

---

> ### Author Response · Authors · 2021-11-21
> **Response to Reviewer UJ5M [2/2]**
>
> > **“The proposed method provides a single principled figure characterizing the entire representation instead of a separate figure for each predicted attribute”. With the proposed linear mapping, “this principled figure” is obtained by simply averaging the edge weights between the concepts and the clusters. I don’t see why linear probing cannot obtain such a figure by averaging the per-category scores.**
>
> We should note that our measure is not the average of edge weights between concepts and clusters, but the mutual information between concepts and quantized representations (clusters), which is quantified via the predictive capability of the reverse probe. This is fundamentally different from averaging class probabilities.
>
> We agree that it is of course _possible_ to compute a single number for forward linear probes by averaging per-category scores. Our point here is that this score then represents simply that: an average performance of individual attributes. In the previous comment, we discussed a possible issue of this approach in interpreting the feature space.
>
> We will revise the writing to improve clarity regarding this difference between linear probes and our approach.
>
> > **“Our measure can be computed much faster than linear probes”. Please elaborate more on this, as at first glance I thought the proposed method is slower as it involves clustering.**
>
> Given a dataset (e.g. ImageNet) and a set of experts (e.g. the ones listed in Table 1), all labels can be pre-computed for all data points. Feature vectors for a given model can be also pre-computed and stored for the whole dataset. Our method computes cluster assignments using the fast K-means implementation of faiss (typically less than 5min for 2048-d vectors on 4 GPUs*). We then train a linear model to predict these from binary input vectors, which converges in less than 100 epochs in a matter of minutes on a single GPU (1 epoch takes 2sec). On the other hand, standard linear probing on ImageNet typically requires full images (including online augmentations) and multiple forward passes through the frozen feature extractor, which makes it significantly slower to train.
>
> *The GPUs used in our experiments are NVIDIA RTX A4000.
>
> > **From the experiments, it is still not obvious to me that the proposed method is better than linear probing. Some observations listed in the paper are as expected or already pointed out by previous literature…**
>
> Our work focuses on the post hoc interpretation of the _raw representations_ that self-supervised pre-training yields, which differs significantly from evaluating downstream task performance. As a result, it provides complementary insights to existing literature and is not meant to replace standard protocols (e.g. linear classification or finetuning on downstream tasks).
>
> > **Deepcluster-v2 performs better as the optimization objective of deepcluster-v2 is pretty similar to the optimization objective of the proposed metric**
>
> Please see our response to reviewer AZhK regarding methods that use clustering mechanisms. We will include this discussion in the paper.
>
> > **The evaluated self-supervised methods have very similar performance so their rankings may be sensitive to the metrics**
>
> The evaluated self-supervised methods have often even more similar performance in terms of linear classification accuracy on ImageNet or transfer to downstream tasks (e.g. VOC detection), so the same argument can be made to question _existing_ protocols, as well as whether methods have started to “overfit” these protocols. As most methods train similar contrastive mechanisms, this is not very surprising. Our approach measures the interpretability of a learned representation instead of downstream task performance which is a conceptually different evaluation. Further, we derive additional insights about these models (lower performant models rely more on low level cues than better methods) that cannot be obtained from down-stream task evaluation or linear probing.

---

> > ### Author Response · Authors · 2021-12-09
> > **Follow-up**
> >
> > Dear reviewer UJ5M,
> >
> > We thank you again for your feedback on our work. We have addressed your concerns and believe that has clarified several misunderstandings. We would appreciate it if you could let us know whether our response has addressed your concerns and provide an updated assessment of our revised paper. We would be also glad to answer any further questions and clarify any remaining concerns. If you are satisfied by our response, we would appreciate it if you could consider raising your score.
> >
> > Thank you.

---

> ### Author Response · Authors · 2021-11-21
> **Response to Reviewer UJ5M [1/2]**
>
> We thank the reviewer for the thorough reading of our submission and constructive feedback, but we think some points might have been misunderstood. Below we address the reviewer’s concerns and kindly ask them to reconsider their rating. In the revision we will also rephrase the claims that the reviewer has pointed out to provide more clarity with respect to standard linear probing and the reasons why we think (a) expanding the concept space and (b) reversing the probing direction may provide additional insights into self-supervised representation learning.
>
> > **“A limitation of this type of approaches is that one can only discover in the representation meanings that are represented in the available data annotations”. I’m afraid both linear probing and the proposed method have this problem, the proposed method relies on pseudo labels generated by the expert models that were trained with data annotations.**
>
> We have not claimed our approach to be annotation-free. In fact, how could one study the alignment of features and human-interpretable concepts _without_ human-provided labels? What we actually meant to point out is the opposite, i.e. our goal is to exploit _additional_ annotations from _diverse_ sources in order to provide a more holistic evaluation metric than the ones that have been, to date, considered specifically in self-supervised representation learning (SSL). For example, the linear classification accuracy metric in SSL only evaluates representations against the 1000 ImageNet categories. However, as these models are trained without supervision, there is little to no reason why their representation vectors should align perfectly with a specific label distribution. It is thus important to augment the label set to provide good coverage over a number of diverse concepts that extend well beyond ImageNet categories.
>
> As stated in the sentence after the one quoted by the reviewer, “in order to maximize the semantic coverage of  the  analysis,  it  is  thus  customary  to  combine  annotations for several types  of  attributes”, e.g. as done for the Broden dataset (Bau et al., 2017). This is true for existing studies, as well as ours, and _it is a necessary step for post-hoc interpretations of features, rather than a limitation._
>
> > **“The representation might be predictive of combinations of attributes, i.e. it might understand the concept of ‘red apple’ without necessarily understanding the individual concepts, e.g. ‘red’ and ‘apple’.” This statement is correct. Unfortunately, neither linear probing nor the proposed method can resolve this problem, as they both are linear functions, so score(‘red apple’) == score(‘red’) + score(‘apple’) . In order to recognize “red apple”, both methods should have high scores on “red” and “apple”. In other words, the claim that the proposed method can better handle the compositions of the concepts is incorrect from the methodology perspective.**
>
> We believe that this is a misunderstanding. We would like to point to **Figure 2** of the paper, which we have presented with exactly this example in mind and the goal of highlighting the difference between our approach and forward linear probes. Forward probes can be trained as binary classifiers for each attribute, mapping a feature vector to the corresponding value of the attribute. Instead, reverse probes map a binary vector that represents a combination of attributes (color, shape) to a categorical cluster in feature space. The decision boundaries shown in the figure are those of the forward linear probes. If _all_ attributes are linearly separable in the representation space (Fig. 2a), then both forward and reverse probes do well at separating the attributes, which may be thought of as identifying the concepts “red”, “blue”, “square” and “circle”.
>
> However, in Fig. 2b this is not the case: here, the color attribute is not linearly separable, highlighting a failure case of the forward probe. Due to the linear inseparability, the color probe ($h_1$) results in chance accuracy (50%). In this case, score(“red square”) == score(“red”) + score(“square”) is low for the forward probe, because score(“red”) is low. However, please note that the clusters are easily interpretable/nameable by a human observer. The linear inseparability of individual attributes does _not_ affect the reverse probe, which will perform well as long as the clusters are meaningful (as is the case here), and this is precisely what we wish to measure. The reverse probe gets a high score on “red square” without requiring the concept “red” to be _also_ encoded in the features. In this case, we can say that the model has discovered the concept of a “red square” without identifying, in isolation, the concept “red”.
>
> In our revision, we will discuss this example in more detail to prevent misunderstandings.

---

### Official Review · Reviewer_phRe · 2021-11-03

**Correctness:** 4
**Technical Novelty And Significance:** 2
**Empirical Novelty And Significance:** 2
**Recommendation:** 5
**Confidence:** 3

**Main Review:**

I enjoyed reading the paper - it is well written and the proposed approach is simple. In general, interpretability or maybe rather explainability of representations is a subject worth studying, and I appreciate the effort that went into experimentation in this work. However, there are also issues that I will describe below that relate to 1) positioning with respect to related work, and 2) the ability for the reader to draw insights from the analysis, among more minor aspects, that would significantly strengthen the work. In the current state I believe the paper is not yet ready for publication at ICLR.

**Related work: topic models**

In essence, the approach proposed here boils down to a linear model that predicts an output class for inputs in the shape of vectors that indicate image attributes (e.g. object presence, texture, etc). While this type of approach might be new in the assessment of semanticity of image representations, it is strikingly similar to classic topic models in NLP (e.g. a model identifying topics in documents represented as bag-of-words). The authors propose a measure to essentially measure the quality of those "topics" (i.e. clusters in this work), which is a well studied area in NLP where it is usually referred to as "topic coherence", e.g. [1]. All sorts of measures have been proposed to measure topic coherence, including those based on (pointwise) mutual information. I would therefore encourage the authors to take a deeper look into this relationship - it may strengthen the paper if the relationship of this approach to other methods can be understood more thoroughly.

**Ability to draw insights**

The authors run plenty of experiments using recent methods to assess their semanticity, which I think should be the focus point of this paper. While I appreciate the effort that went into this, I think how these results are currently presented (is there a better way to present the findings than table 2?), and discussed falls short of what I expected given the focus of this work. While the authors provide some insights w.r.t. to different types of approaches (e.g. clustering vs contrastive) it would be useful to link this to more established taxonomies (see e.g. [2]). Grouping approaches based on some form of taxonomy may make it easier to identify differences. This is an area where this paper could shine and provide actionable insights for the practitioner but at the moment falls short of that expectation.

**Choice of K**

The choice of the number of clusters seems a bit arbitrary to me, particularly as it impacts the (relative) performance of different methods. The authors outline in the appendix (B.2, in particular Fig 7) how BYOL's performance improves a lot with much larger K, essentially being en-par with the best performing methods. Yet in the main text the authors mention specifically that more recent methods (like BYOL) don't perform as well as others - Would the authors draw different conclusions if K was larger? Again, these differences may be alleviated with a more adequate grouping of approaches across experiments. Also in section 4.5 the authors drop K to 500, why?

**Limitations of a linear probe**

The authors outline how the proposed approach allows more complex relationships between image attributes to be captured. Yet they only present results for a linear model (predicting the cluster in representation space). How well would a non-linear model do in this task, which would be able to capture more elaborate relationships between image attributes?

[1] Roder et al. (2015) Exploring the Space of Topic Coherence Measures

[2] Jing et al (2020) Self-supervised Visual Feature Learning with Deep Neural Networks: A Survey

**Summary Of The Paper:**

This paper presents an approach to investigate the semanticity of representations learned via self-supervision applied to images. The approach measures (via mutual information) how well a linear model can map a vector indicating image attributes (e.g. objects, texture, etc) to clusters within the representation space. The authors apply this method to recent methods from the literature and interpret the resulting ranking, aiming to learn more about which methods produce representations that map well to human judgements of similarity.

**Summary Of The Review:**

The paper is well written and the proposed approach is simple. In general, interpretability of representations is a subject worth studying, and I appreciate the effort that went into experimentation in this work. However, there are also issues that relate to 1) positioning with respect to related work, and 2) the ability for the reader to draw insights from the analysis, among other aspects, that would significantly strengthen the work. In the current state I believe the paper is not yet ready for publication at ICLR.

---

> ### Author Response · Authors · 2021-11-21
> **Response to Reviewer phRe [2/2]**
>
> > **Ability to draw insights. Is there a better way to present the findings than table 2? Grouping approaches based on some form of taxonomy may make it easier to identify differences.**
>
> Grouping approaches to improve readability and insights is a great idea, thank you for suggesting this. Unfortunately, the provided survey paper [2] has very little overlap with the methods that we have investigated (i.e. in our case, most are state-of-the-art methods of the year 2020 or more recent).  [2] discusses mostly supervised methods and the only self-supervised methods are based on generative or context-based pretext tasks, which is mostly not applicable here, so we cannot use the proposed taxonomy. Although there is currently no official taxonomy, perhaps the most common categorization of the methods in our investigation is:
>
> 1. **context-based pretext tasks** — typically training via tasks such as colorization or puzzles
> 2. **contrastive learning** — training by contrasting positive examples (different views of the same image) to negative examples (other images)
> 3. **positive-only** — training using only positive examples (also known as teacher-student)
> 4. **clustering-based** — training via cluster assignments or, usually, combining contrastive learning and clustering
>
> However, we note that the majority of current approaches are contrastive with a recent trend towards positive-only methods. Some, like SwAV and PCL combine contrastive learning with clustering. We have updated Table 2 in the paper to highlight the group to which each method belongs.
>
> Regarding drawing insights, as we discuss in the paper, we have observed that clustering-based methods generally produce more interpretable representations. In Table 2, rows within each epoch group are sorted by ascending NMI and it is easy to see that methods which use some form of clustering appear in the last rows (highest NMI). However, we should note that it is hard to use the taxonomy alone to draw conclusive insights, as performance of models is largely dependent on other factors such as the choice of augmentations or, in some cases, number of training epochs.
>
> Another important insight, as also noted by reviewer RSYr, is that models (trained on ImageNet) capture more than just purely semantic labels and more than just ImageNet labels. They capture information about material, textures, scenes, and their combinations. This is true for all models, however we observe that more performant models recover more of the original label distribution and rely less on other lower-level concepts, such as textures (Figure 4, Table 3).
>
>
> > **Choice of K: Would the authors draw different conclusions if K was larger?**
>
> While this is indeed an important point, the goal of Fig. 7 in the Appendix was actually to show that the choice of K does not matter, i.e. the ranking is mostly robust to the choice of K, using some selected methods. Since ImageNet consists of 1000 classes, we found K=1000 to be a sensible choice for the experiments. Nevertheless, for exhaustiveness, we are currently evaluating more methods over a range of K values and will update the figure with the new results.
>
>
> > **Choice of K: In section 4.5 the authors drop K to 500, why?**
>
> Since in Section 4.5 we present experiments on Places-365, which has fewer classes, we drop K to 500 to reflect this. The exact choice of K does not matter, as long as it is used consistently for all methods under evaluation. In fact, although it is possible to determine K using known methods, such as the silhouette score, this could result in a different number of clusters for each model, which would not allow for relative comparisons between models. For completeness, we will include more values of K.
>
> > **Limitations of a linear probe. The authors outline how the proposed approach allows more complex relationships between image attributes to be captured. Yet they only present results for a linear model (predicting the cluster in representation space). How well would a non-linear model do in this task, which would be able to capture more elaborate relationships between image attributes?**
>
> The choice of linear models is _essential_ to interpretability studies. While, technically, it is very much possible to use more complex non-linear models,  it is important to ensure that the probe does not “do the job” instead, i.e. a concept should be captured by the _raw representation_ rather than some complex mapping operating on top of the representation vector, as it would not be possible to determine whether the original model is responsible for the performance or the probe.

---

> > ### Author Response · Authors · 2021-12-09
> > **Follow-up**
> >
> > Dear reviewer phRe,
> >
> > Thank you again for your valuable feedback on our work. Based on your comments, we have added a discussion on topic models in the appendix, categorized methods under a taxonomy, clarified our findings, and included additional experiments with varying K to verify that this choice does not actually lead to different conclusions. We hope that our revised version has addressed your concerns and we would appreciate it if you could engage with us with feedback on the revision. We are also happy to answer any further questions and clarify any remaining concerns. If you are satisfied by our response, we would appreciate it if you could consider raising your score.
> >
> > Thank you.

---

> ### Author Response · Authors · 2021-11-21
> **Response to Reviewer phRe [1/2]**
>
> Thank you for the valuable feedback on our work!
>
> > **Related work: topic models. The authors propose to measure the quality of "topics" (i.e. clusters in this work), which is a well studied area in NLP where it is usually referred to as "topic coherence".**
>
> Thank you for pointing out the analogy to topic models! This is rather interesting. We agree that there is a similarity between topic coherence and the way we evaluate models. However, this analogy breaks down rather quickly. To discuss this, we can directly map the terms of topic models (topics, documents and words) to our scenario (clusters, images and concepts) in this order. The semantic coherence of a topic (cluster) is often measured by the co-occurrence of words (concepts) across documents (images). One immediate difference that arises is that in topic modeling the original space (corpus) is authored by humans and can be thus considered interpretable. Measures of coherence are introduced to understand whether the topics found by a model are also interpretable, i.e. evaluating the model producing the topics. In our case, we do not know the degree to which the original space (representation) is interpretable and this is precisely what we aim to quantify with our method.
>
> Another conceptual difference is that topic models operate on a higher level of abstraction: a topic is a collection of words that may describe a higher-level idea defined by this collection of words. This is sensible as the goal in topic modelling is to discover which words describe higher-level ideas and can be likely grouped together vs. irrelevant words. In our case, this is not necessary, as the expert annotations already define the relevant concepts. A further abstraction is unnecessary and in fact it may even be undesirable because we are interested in the minimum number of concepts that may explain a cluster. For example, a topic of “animals” may be coherent, yet a cluster of “animals” lacks specificity because it may mix a multitude of categories, instead of purely containing a single species.
>
> Finally, the main mechanism in evaluating topic coherence is co-occurrence of words in the general corpus (documents). In our case, this translates to visual concepts co-occuring in the same image and models relationships between concepts in the images/the real world, whereas we are interested in the relationship between clusters and concepts as they have been learned by the model.
>
> If the reviewer was thinking of a different analogy to topic modelling we are open to discuss. In our understanding, the one presented above fits best but shows that both fields have somewhat different goals and mechanisms to measure performance. We are happy to discuss this connection in the appendix.

---

### Official Review · Reviewer_RSYr · 2021-11-04

**Correctness:** 4
**Technical Novelty And Significance:** 3
**Empirical Novelty And Significance:** 4
**Recommendation:** 8
**Confidence:** 4

**Main Review:**

I actually really enjoyed reading this paper and found the contribution to be compelling. My main suggestion is to surface your insights and findings in the abstract and introduction. These sections, currently, only describe your method and not your findings. The abstract and introduction made me ask “so what?”. The findings are buried; some in the appendix!

I wish the choice of K-means as the clustering algorithm was questioned. For example, if you used hierarchical agglomerative clustering, would that lead to clusters where different levels of the hierarchy could represent compositions of children cluster concepts.

What I think is missing:
1) Which concepts are not captured by any of these models?

2) Which models are capable of decoupling individual concepts (identifying each concept in isolation) versus only compositions of concepts?

3) How does interpretability correlate with transfer to other downstream tasks like object detection, segmentation, etc. I know the paper includes a transfer experiment to Places but there could have been a more thorough exploration of transfer learning.


**Summary Of The Paper:**

This paper studies how human-interpretable are the concepts learned within the representation space of self-supervised models. It argues that linear probing methods are unable to identify whether a representation space contains a specific concept because the input might contain multiple confounding concepts (“red apple”) and might encode them in a manner that renders “red” and “apple” individually linearly inseparable. Instead, it proposes a “reverse linear probe” moving from which maps combinations of binary concept labels to the representation space, which is clustered using k-means.

The paper uses this linear probing method to propose a normalized mutual information metric to measure how well human-interpretable concepts are encoded in the representation space. Even though they find that their metric is correlated with linear probe, there are some interesting insights. First, the paper finds that models trained on ImageNet images capture more than just semantic class labels. They capture information about textures, scenes, objects, etc. Second, they found that additional training epochs dont lead to more interpretable representations even though it increases end-task performance. They also identified that OBoW, does really well even with only 200 epochs (though the reasons for why is still left to be investigated). Clustering-based approaches generally produce more interpretable representations. Fourth, ViT is a better architecture. Finally, qualitative evaluation of clusters seem to justify their utility of the reverse probe since NMI increases as when accounting for multiple categories for objects that occur together (ex, french horn and people).


**Summary Of The Review:**

Overall, I am still quite satisfied with where the paper currently stands. I listed things I would like to see improved or discussed. I also listed some potential questions for investigation. But these questions do not detract from my overall positive impression of the paper.

---

> ### Author Response · Authors · 2021-11-21
> **Response to Reviewer RSYr**
>
> We thank the reviewer for the positive and motivating feedback. We really appreciate the suggestion to surface the paper's findings in the abstract/introduction and we have revised the paper accordingly to reflect this.
>
> > **I wish the choice of K-means as the clustering algorithm was questioned. For example, if you used hierarchical agglomerative clustering, would that lead to clusters where different levels of the hierarchy could represent compositions of children cluster concepts.**
>
> We experimented with other clustering algorithms, such as BIRCH, but found K-means to be a reasonable choice. The difficulty in choosing the clustering algorithm is that it is not possible to know which algorithm works best in this case and in absence of ground truth. We really like the idea of hierarchical clustering and are currently experimenting with it. However, besides qualitative findings, we should note that it is not clear how one could use hierarchies for quantitative evaluation, especially since measuring the mutual information over different $K$ yields incomparable results.
>
>
> >  **Which concepts are not captured by any of these models?**
>
> Thank you for suggesting this! It is possible to find concepts that are “well-learned” or “not captured” by examining the linear model’s coefficients and the predictive performance for each cluster. Concepts not captured by most models are certain scene attributes (e.g. “studying”, “research”, “stressful” which may require a certain degree of reasoning), as well as small, less common household objects (e.g. can opener, paper cutter, cooking spray), which are likely not salient regions in a dataset like ImageNet. Clusters with high weights on such categories have a lower prediction score (e.g. NMI or accuracy) and are far less coherent.
>
>
> >  **Which models are capable of decoupling individual concepts (identifying each concept in isolation) versus only compositions of concepts?**
>
> Generally, more performant models are better at decoupling individual concepts; these often also cluster more “pure” IN-1k categories in their representation space (Fig. 4, blue bars). However, we have observed that, even then, _context_ plays a prominent role in the representation (e.g. Fig. 9c: whale in _man-made environment_ vs. whale in the _ocean_, Fig.10c: snake on a _hand_ vs. snake on the _ground_, Fig. 11a: citrus _slices_ vs. the whole fruit). This is true even for models that do well at separating IN-1k categories, i.e. the true underlying label distribution. This likely also explains why in [1] ImageNet self-supervised models do as well as or better than the supervised baseline at context, counting and gestalt tasks. Finally, we should note that for _all_ models, compositions are very common with human-related concepts, such as faces or hands (Fig. 5, Fig. 8c&d, Fig. 10c).
>
> >  **How does interpretability correlate with transfer to other downstream tasks like object detection, segmentation, etc.**
>
> This is indeed a very good point, which we have also considered, but performance reports for transfer tasks are not always consistent across the literature (e.g. evaluation protocols are changing, or different tasks are evaluated); so it has been more difficult to obtain reliable data points to measure the correlation. Upon your suggestion, however, we computed the correlation of interpretability and transfer performance of models on VOC Detection (one of the more standardized tasks) and found it to be more correlated than that of standard linear probing with the transfer task. We will revise the appendix with the relevant plots.
>
> -----
> [1] Van Horn, et al. "Benchmarking Representation Learning for Natural World Image Collections." Proceedings of the IEEE/CVF Conference on Computer Vision and Pattern Recognition. 2021.

---

> > ### Author Response · Authors · 2021-11-23
> > **Note on correlation with downstream tasks**
> >
> > Ultimately, we did not observe any significant significant p-values when measuring correlations with downstream tasks. This is likely due to the fact that (a) sample size is small: we do not have enough data points (models) that report performance on other tasks, (b) often there are incosistencies w.r.t. performance reports from paper to paper, and (c) for example on VOC detection, most methods reach AP50 between 82% and 83% (i.e. just one point difference), while the measured interpretability varies more significantly. Due to these reasons, we refrain from making any conclusive comments for now, but will keep investigating this point.

---

### Author Response · Authors · 2021-11-21
**General response to all reviewers**

We would like to thank all reviewers for their valuable feedback. We are happy that reviewers appreciated the valuable insights [RSYr] and the amount of experimentation and evaluated models [phRe, UJ5M, AZhK]. We also appreciate the constructive comments and suggestions for strengthening our work. Below we respond to the reviewers' comments and we will soon also upload a revision to our submission based on their feedback.

---

### Author Response · Authors · 2021-11-23
**Revision Summary**

We have revised our submission according to the reviewers’ comments; thank you again for the constructive feedback. For convenience, major edits and added content are shown with green font.

Summary of changes:

* Some key insights are summarized in the introduction and more added to the qualitative analysis **[RSYr]**
* We have revised the related work to include [Escorcia et al. 2015] and [Oramas et al. 2019] **[AZhK ]**
* We have added a longer description of Fig. 2 to better highlight the difference between standard linear probes and our approach and what makes reverse probing more suitable for the task we are considering **[UJ5M, phRe]**
* We introduced icons to categorize self-supervised methods as suggested by **[phRe]**
* We discussed the relation to topic models and topic coherence evaluation in the appendix **[phRe]**
* Missing implementation details and computation times are added in the appendix. **[AZhK, UJ5M]**
* We included more comprehensive evaluation over varying K by adding more methods in Fig. 7 and relevant discussion **[phRe]**
* We have specifically discussed self-supervised methods based on clustering in the appendix **[AZhK, UJ5M]**

---

### Decision · Program_Chairs · 2022-01-20

**Decision:**

Accept (Poster)

**Comment:**

This paper proposes a method for inspecting and interpreting the visual representations learned by self-supervised methods.
The method is conceptually simple and intuititive, the authors assume that concept labels for the images are available, and then go on to learn a mapping between the learned image vectors and the human-provided descriptions of the images. The key insight is to learn a reverse mapping, i.e., to map label vectors to representation vectors. Specifically, feature vectors are quantized using k-means to obtain clusters;  images are labeled (automatically) with a diverse set of concepts from expert models trained with supervision on
external data sources, and  a linear model is trained  to map concepts to clusters, measuring the mutual information between the representation and human-interpretable concepts.

Reviewers raised some questions regarding the relation of the approach to topic models, the difference between reverse probing and linear probing, implementation details and computation. The authors addressed reviewers comments convincingly with additional experiments and/or explanations.